# Integrating muti-omics data to identify tissue-specific DNA methylation biomarkers for cancer risk

Yaohua Yang [1,8] ✉, Yaxin Chen[2,8], Shuai Xu [3], Xingyi Guo [3], Guochong Jia [3], Jie Ping [3], Xiang Shu[4], Tianying Zhao [3], Fangcheng Yuan [3], Gang Wang[2], Yufang Xie[2], Hang Ci[2], Hongmo Liu[2], Yawen Qi[2], Yongjun Liu[5], Dan Liu[2], Weimin Li [2], Fei Ye [6], Xiao-Ou Shu[3], Wei Zheng [3], Li Li[7], Qiuyin Cai [3] ✉ & Jirong Long [3] ✉

The relationship between tissue-specific DNA methylation and cancer risk remains inadequately elucidated. Leveraging resources from the Genotype-Tissue Expression consortium, here we develop genetic models to predict DNA methylation at CpG sites across the genome for seven tissues and apply these models to genome-wide association study data of corresponding cancers, namely breast, colorectal, renal cell, lung, ovarian, prostate, and testicular germ cell cancers. At Bonferroni-corrected $P < 0.05$, we identify 4248 CpGs that are significantly associated with cancer risk, of which 95.4% (4052) are specific to a particular cancer type. Notably, 92 CpGs within 55 putative novel loci retain significant associations with cancer risk after conditioning on proximal signals identified by genome-wide association studies. Integrative multi-omics analyses reveal 854 CpG-gene-cancer trios, suggesting that DNA methylation at 309 distinct CpGs might influence cancer risk through regulating the expression of 205 unique *cis*-genes. These findings substantially advance our understanding of the interplay between genetics, epigenetics, and gene expression in cancer etiology.

Cancer is a complex disease with an estimated overall heritability of 33%[1]. Genome-wide association studies (GWAS) have identified over 1000 common variants associated with cancer risk[2–9]. However, these variants are mainly situated in non-coding regions, posing challenges in identifying target genes and mechanisms[2–9]. While expression quantitative trait loci (eQTL) studies have uncovered many genes associated with GWAS-identified variants, most of them were limited by the small proportion of gene expression variation captured by individual eQTL variants[10]. Transcriptome-wide association studies (TWAS) made strides in addressing this by incorporating multiple *cis*-variants to predict gene expression, unveiling hundreds of candidate cancer susceptibility genes[11]. However, over half of GWAS loci lacked TWAS-identified genes, implying the possible existence of additional mechanisms contributing to the

[1]Center for Public Health Genomics, Department of Public Health Sciences, UVA Comprehensive Cancer Center, School of Medicine, University of Virginia, Charlottesville, VA, USA. [2]Institute of Respiratory Health, Frontiers Science Center for Disease-Related Molecular Network, State Key Laboratory of Respiratory Health and Multimorbidity, Department of Respiratory and Critical Care Medicine, West China Hospital, Sichuan University, Chengdu, Sichuan, China. [3]Division of Epidemiology, Department of Medicine, Vanderbilt Epidemiology Center, Vanderbilt-Ingram Cancer Center, Vanderbilt University Medical Center, Nashville, TN, USA. [4]Department of Epidemiology and Biostatistics, Memorial Sloan Kettering Cancer Center, New York, NY, USA. [5]Department of Laboratory Medicine and Pathology, University of Washington Medical Center, Seattle, WA, USA. [6]Department of Biostatistics, Vanderbilt University Medical Center, Nashville, TN, USA. [7]Department of Family Medicine, UVA Comprehensive Cancer Center, School of Medicine, University of Virginia, Charlottesville, VA, USA. [8]These authors contributed equally: Yaohua Yang, Yaxin Chen. ✉e-mail: vta8we@virginia.edu; qiuyin.cai@vumc.org; jirong.long@vumc.org

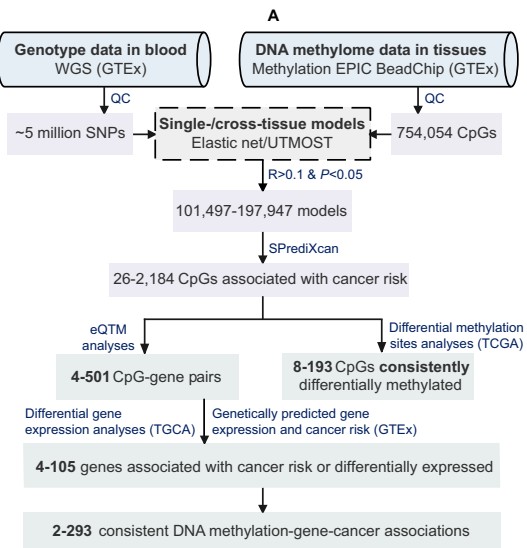

**Fig. 1 | Overall workflow and resources of the present study. A** the overall workflow. The range of values denotes the minimum and maximum numbers across all tissue or cancer types. WGS whole genome sequencing, GTEx Gene-Tissue Expression consortium, QC quality control, UTMOST Unified Test for MOlecular SignaTures, eQTM expression quantitative trait methylation, TCGA The Cancer Genome Atlas. **B** tissue samples used in DNA methylation prediction model development and cancer GWAS data used in association analyses. N number of samples, GWAS genome-wide association studies. The (**B**) was created with BioRender.com released under a Creative Commons Attribution-NonCommercial-NoDerivs 4.0 International license.

genetic susceptibility of cancer risk beyond *cis*-gene expression regulation.

DNA methylation plays crucial roles in regulating gene expression, maintaining genomic stability, and establishing cell identity[12]. Aberrant DNA methylation patterns, such as global hypomethylation and gene-specific hypermethylation, are hallmarks of cancers[13]. In addition to environmental factors, DNA methylation is also shaped by genetics[14]. In the largest GWAS of blood DNA methylation conducted to date ($n = 32,851$), methylation QTLs (mQTLs) were identified for ~45.2% (190,102) of CpGs on the Illumina HumanMethylation450 BeadChip[14]. A recent study employed the Illumina MethylationEPIC BeadChip to profile DNA methylation in 987 samples from 424 subjects, representing nine distinct tissue types, and discovered mQTLs for 37.9% (286,152) of all investigated CpGs. Of these mQTLs, 37% were detected among all tissues while 5% were specific to a particular tissue type[15]. In addition, a subset of these mQTLs were found to colocalize with GWAS-identified loci for various traits in biologically relevant tissues[15]. However, most of these colocalizations did not involve eQTLs, suggesting that genetic effects on trait variations in those loci were more likely to be mediated by DNA methylation rather than gene expression[15]. Therefore, dissecting tissue-specific genetically predicted DNA methylation holds potential in unraveling the genetic susceptibility to complex traits, including cancer susceptibility. We previously discovered 1343 CpGs with genetically predicted DNA methylation levels in blood associated with cancer risk[16–19]. However, the lack of tissue DNA methylation data hindered the evaluation of these findings in cancer-relevant tissues.

Here, we aim to identify tissue-specific DNA methylation biomarkers associated with cancer risk and decipher the underlying mechanisms. Leveraging normal tissue DNA methylation data and paired genetic data of cancer-free donors from the Gene-Tissue Expression (GTEx) consortium, we develop statistical models for predicting DNA methylation at CpGs across the genome for seven tissue types. These models are subsequently applied to cancer GWAS data to infer associations between genetically predicted CpG methylation and the risk of breast, colorectal, renal cell, lung, ovarian, prostate, and testicular germ cell cancers, respectively. For identified cancer-associated CpGs, we employ integrative analyses of DNA methylomic, transcriptomic, genomic, and cancer GWAS data to further explore whether they may affect cancer risk through modulating the expression of nearby genes.

## Results

### Tissue-specific DNA methylation prediction models
The analytical framework of this study is illustrated in Fig. 1. Processed DNA methylation data, including beta mixture quantile (BMIQ)-normalized $\beta$ values of 754,054 CpGs across 987 tissue samples from 424 cancer-free GTEx subjects, were obtained from the Gene Expression Omnibus (GEO). Genotype data was acquired from the database of Genotype and Phenotype (dbGaP). After excluding 131 samples from 57 subjects lacking genetic data, our study retained 856 samples from 367 subjects, primarily of European (86.6%) and African (12.0%) ancestry. These samples comprised 49 breast, 189 colon, 47 kidney, 190 lung, 140 ovary, 105 prostate, 47 testis, 47 whole blood, and 42 muscle tissue samples.

For each CpG site within a specific tissue, we developed prediction models using two approaches, retaining the model with the best predictive performance. For a given set of *cis*-variants of a CpG, both approaches utilize a variable selection algorithm to identify variants contributing to CpG methylation variation and estimate their respective weights in influencing CpG methylation. Briefly, the single-tissue prediction model was constructed using the elastic net method implemented in the R package *glmnet* (v4.1-8)[20], relying exclusively on data from the specific tissue. The cross-tissue prediction model was established employing the multivariate-response penalized regression[21] strategy integrating information from tissues with similar genetic influences on DNA methylation at the CpG site. As expected, prediction performance (*R*-value) of cross-tissue models were significantly higher than that of single-tissue models (*P* from two-sided paired *t*-test of ≈0). Of the 754,054 CpGs investigated, models for 478,360 (63.4%) exhibited reliable prediction performance of $R > 0.10$ and $P < 0.05$. Notably, 46.8% ($n = 224,052$) of these models were highly tissue-specific, found exclusively in one tissue, while only 3.2% ($n = 15,368$) were ubiquitous across all tissues. Specifically, we built

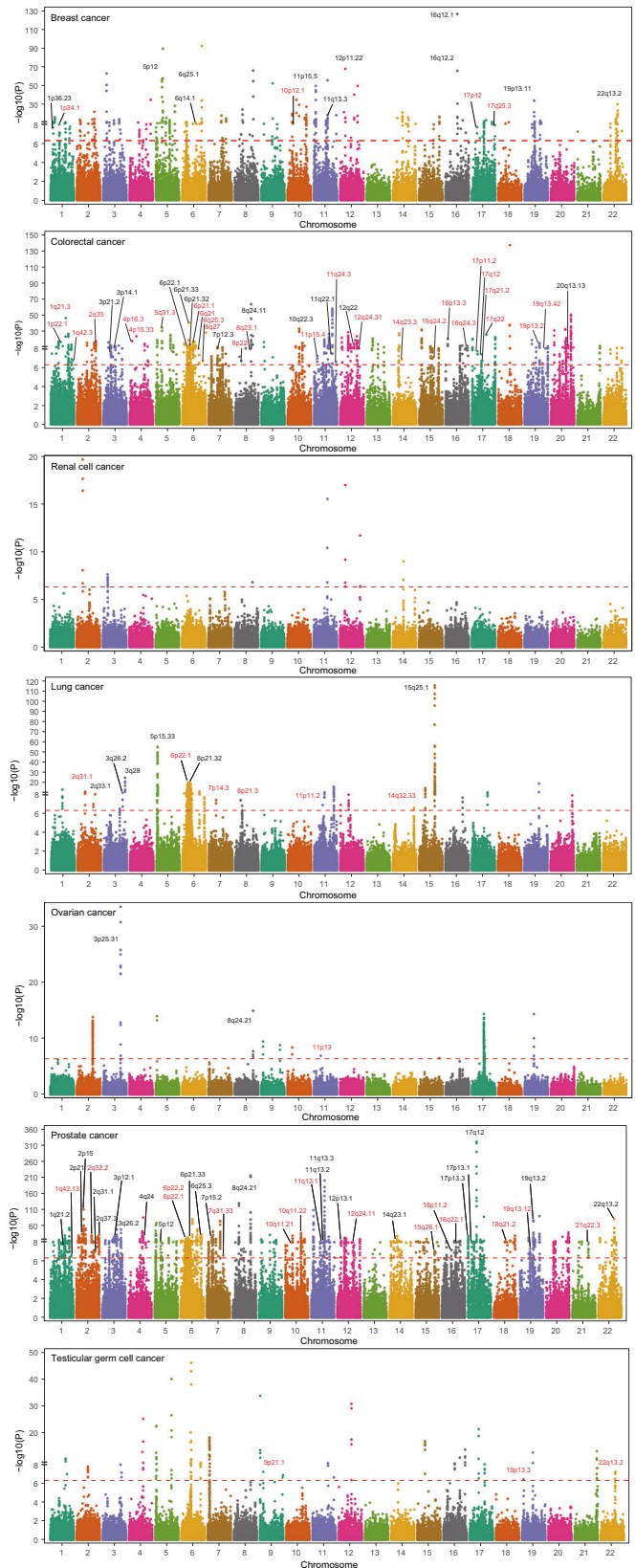

**Fig. 2 | Manhattan plots showing associations between genetically predicted DNA methylation at CpGs and cancer risk.** Association analyses were conducted using SPrediXcan and all statistical tests were two-sided. The x-axis denotes chromosomes and the y-axis is $-\log_{10}P$. The dashed red line in each plot represents the Bonferroni-corrected threshold, which was $4.93 \times 10^{-7}$ for breast cancer, $2.53 \times 10^{-7}$ for colorectal cancer, $3.98 \times 10^{-7}$ for renal cell cancer, $2.55 \times 10^{-7}$ for lung cancer, $2.66 \times 10^{-7}$ for ovarian cancer, $3.28 \times 10^{-7}$ for prostate cancer, and $4.22 \times 10^{-7}$ for testicular germ cell cancer. Loci (cytobands) instead of CpGs are displayed because of the huge number of cancer-associated CpGs. All loci containing CpG-cancer associations that might be independent of GWAS-identified signals are annotated. Potential novel loci are highlighted in red. Source data are provided as a Source Data file.

model established by our method, we attempted to construct a model based solely on the best *cis*-mQTL[22] utilizing the identical data used in our prediction approach. We found that across the seven tissues, the single best *cis*-mQTL method was only able to build reliable models (R > 0.10 and P < 0.05) for an average of 33.1% (interquartile range [IQR]: 24.6%–41.8%) of CpGs with reliable models developed by our strategy. Moreover, For CpGs with models built by both our method and the single best *cis*-mQTL approach, our method consistently outperformed the latter in terms of prediction accuracy (R value), with P of two-sided paired t-test of ≈ 0 (Supplementary Fig. 1).

## Association between genetically predicted DNA methylation and cancer risk

For each tissue type, prediction models were applied to GWAS data of the corresponding cancer using SPrediXcan (v0.7.5)[23] to identify CpGs with genetically predicted DNA methylation levels significantly associated with cancer risk at Bonferroni-corrected two-sided P < 0.05. The Z score of each CpG-cancer association was calculated by summing the Z score of the association between each variant in the CpG's prediction model and cancer risk, each weighted by the variant's effect on CpG as well as the variances of both the variant and the CpG (see "Methods" and ref. [23]). Cancer GWAS data were obtained from multi-ancestry meta-analyses, except for renal cell and testicular germ cell cancers, for which data are only available for European descendants, and the number of cases varied from 10,156 (testicular germ cell cancer) to 158,742 (breast cancer) (see "Methods", Supplementary Data 1 and Fig. 1B). In total, 4248 distinct CpGs were found to be significantly associated with the risk of at least one cancer, yielding 4461 CpG-cancer pairs (Fig. 2 and Supplementary Data 2–8). Remarkably, 95.4% (4052) of these 4248 CpGs showed associations that were exclusive to a specific cancer type (Supplementary Fig. 2). For each CpG-cancer pair, we conducted colocalization analyses using the R package *Coloc* (v5.2.3)[24] to determine whether CpG methylation and cancer risk were influenced by the same causal variant or by distinct variants in linkage disequilibrium (LD). Briefly, utilizing data of variant-CpG and variant-cancer associations for all variants in the CpG's *cis*-region, *Coloc* quantitatively assessed the posterior probabilities for hypotheses about shared genetic causation[24]. Of the 4461 CpG-cancer pairs, 1454 (32.6%) and 866 (19.4%) demonstrated a moderate (PP.H4 > 0.50)[15] to high (PP.H4 > 0.80) probability of colocalization (Supplementary Data 2–8). Notably, of the 248 CpGs significantly associated with ovarian cancer risk, 200 (80.6%) and 117 (47.2%) (117 pairs) exhibited a moderate to high possibility of colocalization with ovarian cancer risk, respectively (Supplementary Data 6).

Among these 4461 CpG-cancer pairs, 4210 CpGs are distributed across 453 of the 801 (56.5%) cancer susceptibility loci identified by previous GWAS[2–9] (Supplementary Data 9–15), while the remaining 251 CpGs at 73 loci are at least one mega base (Mb) away from any GWAS-identified cancer risk variants (Fig. 2). We examined whether these 4461 significant associations might be independent of GWAS signals. In brief, for each CpG-cancer pair, we performed GCTA-COJO (v1.93.2 beta)[25] analyses for each variant in the CpG's prediction model to

101,497 models for breast, 197,947 for colon, 125,745 for kidney, 195,764 for lung, 187,911 for ovary, 152,341 for prostate, and 118,568 for testis tissues, respectively.

We then appraised whether our prediction strategy that leverages multiple *cis*-variants could enhance predictive performance compared to using only the single best *cis*-mQTL. For each CpG with a reliable

obtain the Z score of the variant-cancer association conditioning on all variants that were independently associated with cancer risk in the CpG's nearest GWAS-identified cancer risk locus. The resulting Z scores were used for SPrediXcan[23] analyses to re-assess the CpG-cancer association. We found that 254 of the 4461 associations remained significant at the same Bonferroni-corrected thresholds used in our primary analyses (Supplementary Data 2–8). Among these 254 CpG-cancer pairs, 92 CpGs at 55 loci are >1 Mb away from any GWAS-reported variants, including eight CpGs at four loci for breast cancer, 39 CpGs at 26 loci for colorectal cancer, nine CpG at six loci for lung cancer, one CpG at one locus for ovarian cancer, 28 CpGs at 15 loci for prostate cancer, and seven CpGs at three loci for testicular germ cell cancer (Table 1 and Supplementary Data 2–8), suggesting that these loci might be putative novel cancer susceptibility loci. The remaining 162 CpGs reside in 52 known risk loci, including 14 CpGs at 10 loci for breast cancer, 70 CpGs at 11 loci for colorectal cancer, 10 CpGs at six loci for lung cancer, two CpGs at two loci for ovarian cancer, and 66 CpGs at 23 loci for prostate cancer (Table 2 and Supplementary Data 2–8). In summary, our study identified a substantial number of CpGs whose DNA methylation might mediate the genetic effects on cancer risk in 56.5% of known GWAS loci, revealed 55 putative novel loci, and detected association signals that were independent of GWAS-identified risk variants in 52 known GWAS loci.

We compared the capability of the approach we employed here with the TWAS approach in delineating associations at known cancer susceptibility loci identified by previous GWAS. Results of gene- and splicing-based TWAS were either obtained from recent studies[4,26] or generated by SPrediXcan analyses using GTEx (v8)-based prediction models and cancer GWAS data. Briefly, prediction models for gene expression and splicing variants were developed utilizing genetic and transcriptomic data of cancer-relevant tissue samples primarily from GTEx participants[4]. These models were then applied to cancer GWAS data used in the present study to identify significant associations at Bonferroni-corrected thresholds (see "Methods"). Despite the on average 53.3% (IQR: 30.6%–68.5%) smaller sample size in prediction model development of our study, we detected significant associations signals in more known cancer risk loci than TWAS (453 vs. 377), particularly for renal cell cancer (seven vs. one) and prostate cancer (182 vs. 121) (Supplementary Data 9–15). Noteworthy, in 31.6% of (143 out of 453) known loci containing cancer-associated CpGs identified in the present study, TWAS was unable to identify any significant associations. These results emphasize the effectiveness of our approach in detecting association signals within GWAS-identified loci compared to TWAS.

For cancer-associated CpGs identified in this study, we evaluated their differential DNA methylation between tumor and adjacent normal tissues using data from The Cancer Genome Atlas (TCGA)[27]. Due to the limited availability of DNA methylation data for ovarian and testicular germ cell cancer ($n < 10$), this analysis was restricted to the other five types of cancer. Briefly, for each cancer type, normalized DNA methylation data were used for differential methylation analyses using linear mixed models implemented in the R package *nlme* (v3.1.140), in which tissue status was (tumor/adjacent normal) modeled as a random effect and relevant covariates were adjusted (see "Methods"). Among the 4064 CpGs associated with these five cancers, data on 2142 (52.7%) were available in TCGA, which is consistent with the coverage difference between the DNA methylome profiling arrays used by the present study (Illumina MethylationEPIC array) and TCGA (HumanMethylation450 BeadChip). Of these 2142 CpGs, 469 (21.9%) showed differential DNA methylation at two-sided $P < 0.05$ with directions of effects consistent with those of CpG-cancer associations (Supplementary Data 16–20). This included 82 out of 221 CpGs for breast cancer, 86 out of 357 CpGs for colorectal cancer, eight out of 13 CpGs for renal cell cancer, 100 out of 382 CpGs for lung cancer, and 193 out of 1169 CpGs for prostate cancer.

## DNA methylation influencing cancer risk through modulating *cis*-gene expression

To search for potential target genes of cancer-associated CpGs, we performed expression quantitative trait methylation (eQTM) analyses using GTEx data of the corresponding tissue. To accurately estimate the association between DNA methylation and gene expression, directly measured DNA methylation and gene expression data were used. Each CpGs was examined for its DNA methylation level in association with expression level of each gene within its *cis*-region (500 Kilobase [Kb]) via linear regression analyses. At the false discovery rate (FDR)-corrected two-sided $P < 0.05$, we identified 1369 CpG-gene association pairs, including 251 (62 CpGs and 61 genes) in breast, 58 (45 CpGs and 49 genes) in colon, four (four CpGs and four genes) in kidney, 391 (103 CpGs and 68 genes) in lung, 501 (84 CpGs and 38 genes) in ovary, 151 (115 CpGs and 127 genes) in prostate, and 13 (12 CpGs and 13 genes) in testis. Genes involved in these CpG-gene pairs were then assessed for their impacts on the proliferation of relevant cancer cells using essentiality screening data from DepMap[28]. Briefly, the DepMap project carried out genome-wide Clustered Regularly Interspaced Short Palindromic Repeat (CRISPR)-Cas9 knockout experiments to assess gene essentiality in cellular survival and proliferation, measured by CERES scores, across nearly 1100 cell lines encompassing over 70 cancer types and subtypes[28]. Out of 180 genes with data available in DepMap, 33 demonstrated essential roles in cell proliferation at a commonly used threshold of median CERES Score <-0.50[28], including *CDC20*, *NSF*, *ELL*, and *CAPZB* in breast cancer cells, *CCND1* for renal cell cancer cells, *DHX16*, *HYOU1*, *ABCF1*, *PPP1R10*, *NOL11*, *BPTF*, *DDX39B*, *EXOC3*, and *VARS2* for lung cancer cells, *CDC27*, *NSF*, and *KANSL1* for ovarian cancer cells, and *LSM2*, *CTDP1*, *LSM6*, *ABCE1*, *TCP1*, *PFDN6*, *MZT1*, *CPSF3*, *DDX23*, *GTF2H4*, *MRPL45*, *SMG7*, *NOLC1*, *FDPS*, *C1QTNF4*, and *PRRC2A* in prostate cancer cells (Supplementary Fig. 3).

For the 360 genes significantly associated with cancer-associated CpGs in eQTM analyses, we conducted SPrediXcan[23] analyses to investigate the genetically predicted expression levels of them in association with cancer risk utilizing GTEx-based gene expression models built by previous studies[29,30] and cancer GWAS data used in the present study. We also acquired data on cancer-normal differential expression of these genes from the Gene Expression Profiling Interactive Analysis (GEPIA2) web server[31], in which gene expression data from TCGA and GTEx were jointly analyzed using a uniform pipeline and differential expression analyses were performed using the R package *limma* (v3.18)[31]. We found that 55 of 61 (90.2%) genes for breast cancer, 46 of 49 (93.9%) genes for colorectal cancer, four of four (100.0%) genes for renal cell cancer, 64 of 68 (94.1%) genes for lung cancer, 31 of 38 (81.6%) genes for ovarian cancer, 105 of 127 (82.7%) genes for prostate cancer, and 13 of 13 (100.0%) genes for testicular germ cell cancer showed a significant association or differential expression at FDR-corrected two-sided $P < 0.05$. By integrating findings from associations between CpGs and cancers, between CpGs and genes, and between genes and cancers, we revealed 854 CpG-gene-cancer trios. Within each trio, the relationships of CpG-cancer, CpG-gene, and gene-cancer showed consistent directions (Table 3 and Supplementary Data 21–27). Involved in these 854 trios were 309 unique CpGs, 205 distinct *cis*-genes of these CpGs, and seven cancers, suggesting that DNA methylation at these CpGs influencing cancer risk through regulating the expression of these genes. Specifically, there were 167 trios (54 CpGs and 38 genes) for breast cancer, 39 (30 CpGs and 34 genes) for colorectal cancer, two (two CpGs and two genes) for renal cell cancer, 293 (89 CpGs and 50 genes) for lung cancer, 273 (81 CpGs and 21 genes) for ovarian cancer, 72 (66 CpGs and 63 genes) for prostate cancer, and eight (eight CpGs and eight genes) for testicular germ cell cancer (Supplementary Data 21–27), respectively. For example, as shown in Table 3 and Fig. 3A,

**Table 1 | Selected[a] DNA methylation marks significantly associated with cancer risk identified in genomic regions not yet reported for cancer risk**

| CpG | Cytoband | Closest gene | Classification | Z[b] | OR (95% CI)[b] | P[b] | R[c] | Nearest GWAS variants | Distance (Mb) | P[b] adjusted for GWAS variants | Coloc PP.H4 |
|---|---|---|---|---|---|---|---|---|---|---|---|
| **Breast cancer** | | | | | | | | | | | |
| cg22991963 | 1p34.1 | ERI3 | Intronic | -5.75 | 0.76 (0.70-0.84) | $8.93 \times 10^{-9}$ | 0.32 | rs3790585 | 1.2 | $4.93 \times 10^{-8}$ | 0.86 |
| cg08856118 | 10p12.1 | ANKRD26 | TSS1500 | -6.03 | 0.77 (0.70-0.84) | $1.59 \times 10^{-9}$ | 0.31 | rs541079479 | 4.5 | $9.39 \times 10^{-10}$ | 0.18 |
| cg22652561 | 17p12 | MAP2K4 | TSS200 | 5.35 | 1.11 (1.07-1.16) | $8.83 \times 10^{-8}$ | 0.56 | rs78378222 | 4.4 | $7.88 \times 10^{-8}$ | 0.76 |
| cg14173033 | 17q25.3 | FOXK2 | Intronic | -5.07 | 0.71 (0.63-0.81) | $3.94 \times 10^{-7}$ | 0.48 | rs745570 | 2.8 | $2.60 \times 10^{-7}$ | 0.67 |
| **Colorectal cancer** | | | | | | | | | | | |
| cg18755616 | 1q21.3 | KCNN3 | Intronic | 14.39 | 1.71 (1.59-1.84) | $5.95 \times 10^{-47}$ | 0.63 | rs5028523 | -18.1 | $8.23 \times 10^{-47}$ | 1.00 |
| cg27491224 | 4p16.3 | CFAP99 | Intronic | 8.74 | 1.12 (1.09-1.15) | $2.41 \times 10^{-18}$ | 0.48 | rs280097 | -92.5 | $2.48 \times 10^{-18}$ | 1.00 |
| cg20556304 | 6q27 | FAM120B; PSMB1 | Intergenic | 11.09 | 1.36 (1.29-1.43) | $1.34 \times 10^{-28}$ | 0.29 | rs151127921 | 36.8 | $1.57 \times 10^{-28}$ | 0.98 |
| cg04020319 | 12q24.31 | C12orf43 | Intronic | 5.85 | 1.07 (1.05-1.10) | $5.03 \times 10^{-9}$ | 0.51 | rs73208120 | 3.7 | $6.96 \times 10^{-9}$ | 0.99 |
| cg07716131 | 16q24.3 | PABPN1L; CBFA2T3 | intergenic | -5.98 | 0.91 (0.89-0.94) | $2.18 \times 10^{-9}$ | 0.39 | rs62042090 | 2.2 | $1.46 \times 10^{-11}$ | 0.99 |
| **Lung cancer** | | | | | | | | | | | |
| cg21583565 | 2q31.1 | CDCA7 | TSS1500 | -5.21 | 0.83 (0.78-0.89) | $1.89 \times 10^{-7}$ | 0.29 | rs13389804 | 10.1 | $1.93 \times 10^{-7}$ | 0.95 |
| cg24999568 | 7p14.3 | PDE1C | TSS1500 | -5.50 | 0.69 (0.60-0.78) | $3.84 \times 10^{-8}$ | 0.24 | rs79228924 | 10.9 | $3.95 \times 10^{-8}$ | 0.75 |
| cg27501386 | 8p21.3 | XPO7 | Intronic | 5.50 | 1.10 (1.06-1.14) | $3.90 \times 10^{-8}$ | 0.27 | rs11780471 | 5.5 | $2.28 \times 10^{-8}$ | 0.66 |
| cg00417304 | 11p11.2 | PRDM11 | Intronic | 5.29 | 1.94 (1.52-2.49) | $1.20 \times 10^{-7}$ | 0.29 | rs174559 | 16.5 | $1.15 \times 10^{-7}$ | 0.99 |
| cg02390981 | 14q32.33 | KLC1 | Intronic | -5.16 | 0.89 (0.86-0.93) | $2.51 \times 10^{-7}$ | 0.43 | rs1200399 | 68.9 | $2.52 \times 10^{-7}$ | 0.97 |
| **Ovarian cancer** | | | | | | | | | | | |
| cg09087803 | 11p13 | RCN1 | Intronic | 5.30 | 1.25 (1.15-1.36) | $1.18 \times 10^{-7}$ | 0.20 | rs7937840 | 29.8 | $1.13 \times 10^{-7}$ | 0.74 |
| **Prostate cancer** | | | | | | | | | | | |
| cg26452081 | 1q42.13 | WNT9A | Intronic | -5.18 | 0.55 (0.43-0.69) | $2.21 \times 10^{-7}$ | 0.36 | rs1294247 | 5.4 | $1.39 \times 10^{-7}$ | 0.95 |
| cg06524124 | 2q32.2 | NAB1; GLS | Intergenic | 5.14 | 1.03 (1.02-1.04) | $2.69 \times 10^{-7}$ | 0.71 | rs67228975 | 10.3 | $2.56 \times 10^{-7}$ | 0.95 |
| cg10179363 | 15q26.1 | ANPEP | Intronic | 5.23 | 1.04 (1.02-1.05) | $1.71 \times 10^{-7}$ | 0.72 | rs79548680 | 1.2 | $2.70 \times 10^{-7}$ | 0.97 |
| cg23711434 | 18q21.2 | TCF4; LINC01415 | Intergenic | 5.62 | 1.12 (1.08-1.17) | $1.90 \times 10^{-8}$ | 0.44 | rs12955457 | 1.6 | $1.19 \times 10^{-9}$ | 0.97 |
| cg11988872 | 21q22.3 | AGPAT3 | 5'UTR | 5.89 | 1.04 (1.03-1.05) | $3.92 \times 10^{-9}$ | 0.69 | rs9984523 | 2.5 | $2.97 \times 10^{-8}$ | 0.99 |
| **Testis germ cell cancer** | | | | | | | | | | | |
| cg22667787 | 9p21.1 | APTX | TSS200 | -5.45 | 0.60 (0.50-0.72) | $5.05 \times 10^{-8}$ | 0.32 | rs55873183 | 32.1 | $5.12 \times 10^{-8}$ | 0.17 |
| cg23837289 | 19p13.3 | REXO1 | Intronic | 5.11 | 1.36 (1.21-1.53) | $3.22 \times 10^{-7}$ | 0.46 | rs548059 | 20.4 | $3.28 \times 10^{-7}$ | 0.81 |
| cg11677105 | 22q13.2 | SNU13 | Intronic | 5.45 | 1.56 (1.33-1.82) | $5.01 \times 10^{-8}$ | 0.47 | rs739525 | 20.7 | $5.10 \times 10^{-8}$ | 0.68 |

OR odds ratio per standard deviation (SD) increase in genetically predicted DNA methylation levels, CI confidence interval, GWAS genome-wide association studies, Mb mega base, TSS transcription start site, UTR untranslated region, PP.H4 posterior probability that both the CpG site and cancer risk are associated and share a single causal variant.

[a] Of the 4461 significant CpG-associations identified in the present study, 254 remained significant after adjusting for the nearest GWAS-identified variants, in which 92 CpGs are located in 55 putative novel loci that are >1 Mb away from nearest GWAS-identified cancer risk variants. Of these 55 loci, for each cancer type, at most five CpG-cancer pairs showing the highest possibility of colocalization (PP.H4) are presented, and all the other pairs are available in Supplementary Data 2–8. The Bonferroni-corrected significance threshold was two-sided P-value of 4.93 × 10⁻⁷ for breast cancer, 2.53 × 10⁻⁷ for colorectal cancer, 2.55 × 10⁻⁷ for lung cancer, 2.66 × 10⁻⁷ for renal cell cancer, 3.98 × 10⁻⁷ for ovarian cancer, 3.28 × 10⁻⁷ for prostate cancer, and 4.22 × 10⁻⁷ for testicular germ cell cancer.

[b] Association Z scores, ORs, 95% CIs, and P-values were evaluated by applying GTEx-based tissue DNA methylation prediction models to cancer GWAS data using SPrediXcan.

[c] Coefficients of correlation between measured and genetically predicted DNA methylation levels.

**Table 2 | Selected[a] DNA methylation marks significantly associated with cancer risk identified in genomic regions within 1 Mb of known cancer risk variants but representing independent association signals**

| CpG | Cytoband | Closest gene | Classification | Z[a] | OR (95% CI)[a] | P[a] | R[b] | Nearest GWAS variants | Distance (Mb) | P[a] adjusted for GWAS variants | Coloc PP.H4 |
|---|---|---|---|---|---|---|---|---|---|---|---|
| **Breast cancer** | | | | | | | | | | | |
| cg09938876 | 1p36.23 | UTS2 | TSS1500 | 5.60 | 1.22 (1.14-1.31) | $2.10 \times 10^{-8}$ | 0.34 | rs707475 | -0.003 | $3.96 \times 10^{-9}$ | 0.14 |
| cg11557886 | 5p12 | MRPS30 | Intronic | -15.58 | 0.63 (0.59-0.66) | $9.64 \times 10^{-55}$ | 0.45 | rs10941679 | 0.105 | $8.49 \times 10^{-10}$ | 0.29 |
| cg25021969 | 6p14.1 | BCKDHB; TENT5A | Intergenic | -5.99 | 0.76 (0.69-0.83) | $2.15 \times 10^{-9}$ | 0.29 | rs12207986 | 0.081 | $1.24 \times 10^{-7}$ | 0.17 |
| cg24428144 | 12p11.22 | PTHLH; LOC729291 | Intergenic | -17.48 | 0.73 (0.70-0.75) | $2.17 \times 10^{-68}$ | 0.41 | rs7297051 | 0.006 | $1.81 \times 10^{-7}$ | 0.13 |
| cg11385249 | 19p13.11 | MYO9B | Intronic | -5.24 | 0.88 (0.84-0.92) | $1.58 \times 10^{-7}$ | 0.45 | rs67397200 | 0.186 | $2.19 \times 10^{-7}$ | 0.63 |
| **Colorectal cancer** | | | | | | | | | | | |
| cg19593490 | 6p22.1 | HCG9 | Exonic | -5.52 | 0.90 (0.87-0.94) | $3.38 \times 10^{-8}$ | 0.68 | rs1476570 | 0.133 | $2.70 \times 10^{-42}$ | 0.99 |
| cg13115165 | 10q22.3 | ZMIZ1 | Intronic | -5.99 | 0.63 (0.54-0.73) | $2.07 \times 10^{-9}$ | 0.35 | rs1782645 | -0.003 | $1.15 \times 10^{-7}$ | 0.99 |
| cg25472918 | 11q22.1 | TRPC6; ANGPTL5 | Intergenic | 5.52 | 1.07 (1.04-1.09) | $3.30 \times 10^{-8}$ | 0.42 | rs2155065 | 0.015 | $2.48 \times 10^{-8}$ | 0.88 |
| cg12149513 | 12q22 | TMCC3; MIR492 | Intergenic | 5.91 | 1.10 (1.06-1.13) | $3.50 \times 10^{-9}$ | 0.50 | rs1110875 | -0.861 | $2.33 \times 10^{-42}$ | 1.00 |
| cg27204212 | 20q13.13 | LINCO1271; PTPN1 | Intergenic | 9.67 | 2.55 (2.11-3.08) | $4.08 \times 10^{-22}$ | 0.37 | rs6095946 | 0.002 | $1.25 \times 10^{-10}$ | 0.98 |
| **Lung cancer** | | | | | | | | | | | |
| cg23681745 | 2q33.1 | CFLAR | Intronic | 5.80 | 1.09 (1.06-1.13) | $6.76 \times 10^{-9}$ | 0.31 | rs3769821 | -0.119 | $7.06 \times 10^{-9}$ | 0.96 |
| cg21687591 | 3q26.2 | MYNN | Intronic | 6.38 | 1.11 (1.07-1.15) | $1.76 \times 10^{-10}$ | 0.35 | rs2293607 | 0.010 | $1.68 \times 10^{-10}$ | 0.99 |
| cg22662645 | 3q28 | TP63 | TSS200 | 9.51 | 1.70 (1.52-1.90) | $1.99 \times 10^{-21}$ | 0.17 | rs13314271 | 0.008 | $6.09 \times 10^{-10}$ | 0.21 |
| cg25518571 | 5p15 | TERT | Intronic | -14.19 | 0.59 (0.54-0.63) | $1.03 \times 10^{-45}$ | 0.27 | rs7705526 | 0.000 | $2.51 \times 10^{-8}$ | 0.19 |
| cg08293075 | 6p21.32 | PBX2 | Intronic | 5.89 | 1.28 (1.18-1.40) | $3.96 \times 10^{-9}$ | 0.26 | rs34102154 | 0.416 | $4.59 \times 10^{-9}$ | 0.11 |
| **Ovarian cancer** | | | | | | | | | | | |
| cg26405475 | 3q25.31 | SSR3; TIPARP-AS1 | Intergenic | -7.41 | 0.80 (0.75-0.85) | $1.27 \times 10^{-13}$ | 0.24 | rs62274041 | 0.112 | $1.89 \times 10^{-7}$ | 0.22 |
| cg16467921 | 8q24.21 | MYC; PVT1 | Intergenic | -5.38 | 0.69 (0.60-0.79) | $7.25 \times 10^{-8}$ | 0.47 | rs9886651 | 0.017 | $2.13 \times 10^{-7}$ | 0.98 |
| **Prostate cancer** | | | | | | | | | | | |
| cg27467234 | 3q26.2 | LRRC34 | TSS1500 | -7.89 | 0.91 (0.89-0.93) | $3.04 \times 10^{-15}$ | 0.40 | rs2293607 | 0.049 | $1.76 \times 10^{-9}$ | 0.99 |
| cg12473775 | 11q13.2 | RHOD | Intronic | -5.82 | 0.63 (0.53-0.73) | $5.97 \times 10^{-9}$ | 0.54 | rs11227678 | 0.018 | $2.03 \times 10^{-7}$ | 0.75 |
| cg19252671 | 12p13.1 | CDKN1B | Exonic | -6.08 | 0.84 (0.80-0.89) | $1.18 \times 10^{-9}$ | 0.34 | rs2066827 | 0.000 | $5.29 \times 10^{-8}$ | 0.94 |
| cg15958104 | 17p13.3 | RPH3AL | 5'UTR | -5.46 | 0.86 (0.82-0.91) | $4.74 \times 10^{-8}$ | 0.61 | rs684232 | 0.420 | $7.62 \times 10^{-12}$ | 1.00 |
| cg04105270 | 17q12 | HNF1B | TSS1500 | -38.18 | 0.55 (0.53-0.57) | $5.69 \times 10^{-319}$ | 0.61 | rs11263763 | 0.003 | $3.41 \times 10^{-9}$ | 0.99 |

*OR* odds ratio per standard deviation (SD) increase in genetically predicted DNA methylation levels, *CI* confidence interval, *GWAS* genome-wide association studies, *Mb* mega base, *TSS* transcription start site, *UTR* untranslated region, *PP.H4* posterior probability that both the CpG site and cancer risk are associated and share a single causal variant.

[a] Of the 4461 significant CpG-associations identified in the present study, 254 remained significant after adjusting for the nearest GWAS-identified variants, in which 162 CpGs are located in 52 loci that are < 1 Mb away from nearest GWAS-identified cancer risk variants. Of these 52 loci, for each cancer type, at most five CpG-cancer pairs showing the highest possibility of colocalization (PP.H4) are presented, and all the other pairs are available in Supplementary Data 2–8. The Bonferroni-corrected significance threshold was two-sided P-value of $4.93 \times 10^{-7}$ for breast cancer, $2.53 \times 10^{-7}$ for colorectal cancer, $3.98 \times 10^{-7}$ for renal cell cancer, $2.55 \times 10^{-7}$ for lung cancer, $2.66 \times 10^{-7}$ for ovarian cancer, $3.28 \times 10^{-7}$ for prostate cancer, and $4.22 \times 10^{-7}$ for testicular germ cell cancer.

[b] Association Z scores, ORs, 95% CIs, and P-values were evaluated by applying GTEx-based tissue DNA methylation prediction models to cancer GWAS data using SPrediXcan.

[c] Coefficients of correlation between measured and genetically predicted DNA methylation levels.

**Table 3 | Selected[a] CpG-gene-cancer trios suggesting DNA methylation influencing cancer risk by regulating gene expression**

| CpG | Chr | Pos (HG19) | Gene | Distance (Mb) | Cytoband | CpG-cancer | | CpG-gene | | Gene-cancer | |
|---|---|---|---|---|---|---|---|---|---|---|---|
| | | | | | | Dir | $P^b$ | Dir | $P^c$ | Dir | $P^d$ |
| **Breast cancer** | | | | | | | | | | | |
| cg23231268 | 3 | 46,792,462 | CCR9 | 0.848 | 3p21.31 | + | $1.43 \times 10^{-8}$ | - | $1.15 \times 10^{-4}$ | - | $7.36 \times 10^{-8}$ |
| cg18035979 | 5 | 81,575,199 | ATP6AP1L | 0.000 | 5q14.2 | + | $1.21 \times 10^{-9}$ | - | $3.05 \times 10^{-5}$ | - | **$1.95 \times 10^{-9}$** |
| cg14587961 | 7 | 99,991,523 | PILRA | Body | 7q22.1 | + | $3.50 \times 10^{-7}$ | + | $3.70 \times 10^{-5}$ | + | **$1.45 \times 10^{-6}$** |
| cg07546779 | 8 | 29,495,175 | LEPROTL1 | -0.458 | 8p12 | - | $3.54 \times 10^{-13}$ | - | $6.25 \times 10^{-4}$ | + | $4.97 \times 10^{-45}$ |
| cg02301815 | 17 | 44,249,491 | KANSL1-AS1 | -0.021 | 17q21.31 | + | $1.74 \times 10^{-8}$ | - | $2.39 \times 10^{-6}$ | - | **$6.00 \times 10^{-11}$** |
| **Colorectal cancer** | | | | | | | | | | | |
| cg20019365 | 2 | 219,134,978 | RP11-378A13.1 | 0.013 | 2q35 | + | $5.06 \times 10^{-13}$ | + | $7.41 \times 10^{-5}$ | + | **$4.27 \times 10^{-19}$** |
| cg14130039 | 6 | 32,121,225 | HLA-DPA1 | -0.911 | 6p21.32 | - | $4.04 \times 10^{-10}$ | - | $3.82 \times 10^{-4}$ | + | **0.01** |
| cg12934461 | 15 | 90,792,652 | MAN2A2 | -0.653 | 15q26.1 | + | $9.66 \times 10^{-9}$ | - | $6.27 \times 10^{-4}$ | - | $5.05 \times 10^{-130}$ |
| cg19877683 | 17 | 80,969,515 | FN3KRP | 0.281 | 17q25.3 | - | $7.53 \times 10^{-8}$ | - | $1.87 \times 10^{-5}$ | + | $8.10 \times 10^{-25}$ |
| cg19133199 | 19 | 41,869,409 | B9D2 | Body | 19q13.2 | + | $2.48 \times 10^{-15}$ | - | $3.14 \times 10^{-5}$ | - | **$3.08 \times 10^{-10}$** |
| **Renal cell cancer** | | | | | | | | | | | |
| cg13524857 | 11 | 69,240,192 | CCND1 | 0.216 | 11q13.3 | + | $1.61 \times 10^{-7}$ | + | $3.61 \times 10^{-3}$ | + | $6.11 \times 10^{-62}$ |
| **Lung cancer** | | | | | | | | | | | |
| cg09476067 | 6 | 30,418,581 | TRIM39 | 0.107 | 6p21.33 | - | $1.42 \times 10^{-19}$ | + | $1.51 \times 10^{-4}$ | - | $3.29 \times 10^{-17}$ |
| cg15732223 | 11 | 118,551,206 | TREH | 0.001 | 11q23.3 | + | $3.46 \times 10^{-8}$ | + | $5.44 \times 10^{-4}$ | + | **$3.18 \times 10^{-6}$** |
| cg05651442 | 12 | 52,347,030 | KRT2 | -0.691 | 12q13.13 | + | $7.55 \times 10^{-8}$ | - | $7.78 \times 10^{-4}$ | - | $8.82 \times 10^{-79}$ |
| cg22563815 | 15 | 78,856,949 | CHRNA3 | -0.028 | 15q25 | - | $1.02 \times 10^{-26}$ | + | $4.08 \times 10^{-5}$ | - | **$3.91 \times 10^{-35}$** |
| cg26812862 | 17 | 66,011,719 | HELZ | 0.770 | 17q24.3 | + | $1.20 \times 10^{-9}$ | - | $1.18 \times 10^{-3}$ | - | $6.89 \times 10^{-27}$ |
| **Ovarian cancer** | | | | | | | | | | | |
| cg18750960 | 2 | 177,016,417 | HOXD4 | Body | 2q31.1 | - | $8.85 \times 10^{-11}$ | - | $3.74 \times 10^{-3}$ | + | **$8.65 \times 10^{-8}$** |
| cg09087803 | 11 | 32,125,186 | QSER1 | -0.790 | 11p13 | + | $1.18 \times 10^{-7}$ | - | $1.97 \times 10^{-3}$ | - | $1.99 \times 10^{-17}$ |
| cg17117718 | 17 | 43,663,208 | LRRC37A4P | 0.036 | 17q21.31 | + | $2.37 \times 10^{-10}$ | - | $1.28 \times 10^{-12}$ | - | $1.13 \times 10^{-13}$ |
| **Prostate cancer** | | | | | | | | | | | |
| cg24838316 | 6 | 29,895,260 | ZFP57 | 0.246 | 6p22.1 | - | $6.31 \times 10^{-11}$ | + | $1.49 \times 10^{-4}$ | - | **$5.41 \times 10^{-6}$** |
| cg16237302 | 11 | 47,429,196 | ARFGAP2 | 0.231 | 11p11.2 | + | $1.11 \times 10^{-8}$ | - | $3.59 \times 10^{-4}$ | - | **$5.15 \times 10^{-4}$** |
| cg00524169 | 19 | 39,138,063 | SAMD4B | -0.695 | 19q13.2 | - | $3.24 \times 10^{-9}$ | + | $2.18 \times 10^{-4}$ | - | $1.09 \times 10^{-16}$ |
| cg15272956 | 20 | 62,332,704 | RTEL1 | 0.005 | 20q13.33 | + | $1.73 \times 10^{-23}$ | + | $4.23 \times 10^{-4}$ | + | **$6.13 \times 10^{-41}$** |
| cg05092891 | 21 | 37,535,885 | MORC3 | -0.170 | 21q22.12 | - | $3.65 \times 10^{-8}$ | + | $5.70 \times 10^{-6}$ | - | $1.34 \times 10^{-31}$ |
| **Testicular germ cell cancer** | | | | | | | | | | | |
| cg23581489 | 6 | 33,164,210 | B3GALT4 | -0.081 | 6p21.32 | + | $2.40 \times 10^{-9}$ | - | $1.99 \times 10^{-4}$ | - | $4.43 \times 10^{-97}$ |
| cg22340370 | 7 | 2,019,882 | MRM2 | -0.254 | 7p22.3 | + | $2.21 \times 10^{-16}$ | + | $1.07 \times 10^{-3}$ | + | **$5.29 \times 10^{-6}$** |
| cg13353244 | 16 | 50,099,780 | BRD7 | -0.248 | 16q12.1 | - | $2.12 \times 10^{-8}$ | + | $4.69 \times 10^{-4}$ | - | **0.02** |
| cg04198914 | 17 | 36,106,025 | C17orf78 | 0.353 | 17q21.2 | - | $1.12 \times 10^{-19}$ | + | $4.79 \times 10^{-4}$ | - | $2.79 \times 10^{-77}$ |

*Chr* chromosome, *Mb* mega base, *Dir* association direction.

[a] Selected from 854 CpG-gene-cancer trios demonstrating consistent directions of CpG-cancer, CpG-gene, and gene-cancer associations. In each trio, all of the three associations were statistically significant. Due to the large number of such trios, for each cancer, at most five trios in distinct loci are presented and all the other trois are available in Supplementary Data 21–27.

[b] P-values of associations between genetically predicted DNA methylation and cancer risk evaluated by applying GTEx-based DNA methylation prediction models to cancer GWAS data using SPrediXcan. Associations with Bonferroni-corrected P < 0.05 were considered significant.

[c] P-values of associations between tissue DNA methylation and gene expression calculated by linear regression using GTEx data. Associations with false discovery rate (FDR)-corrected P < 0.05 were considered significant.

[d] P-values of (1) associations between genetically predicted gene expression and cancer risk evaluated by applying GTEx-based gene expression prediction models to cancer GWAS data using SPrediXcan, or (2) differential gene expression between cancer and normal tissues obtained from GEPIA2. Associations or differential expressions with FDR-corrected P < 0.05 were considered significant. For genes with both P-values available, the one from SPrediXcan analysis is presented. All P-values from SPrediXcan analyses are highlighted in bold.

genetically predicted DNA methylation at cg02301815 was associated with increased breast cancer risk, which was consistent with the higher methylation of this CpG in breast cancer tissues than in adjacent normal tissues. This association may be explained by the negative association between DNA methylation at cg22872885 and expression of the *KANSL1-AS1* gene, and the negative association between genetically predicted expression of *KANSL1-AS1* and breast cancer risk. Consistently, the expression of *KANSL1-AS1* was significantly higher in breast cancer tissues than in adjacent normal tissues. Such examples for other cancers are illustrated in Supplementary Fig. 4.

## Discussion

In this comprehensive investigation of tissue-specific DNA methylation and cancer risk using genetic instruments, we identified 4248 CpGs with predicted normal tissue DNA methylation levels significantly associated with cancer risk, with >95% being specific to one out of seven particular cancer types. Systematic analyses of multi-omics data revealed 854 CpG-gene-cancer trios indicating that 309 unique CpGs may influence cancer risk via modulating the expression of 205 distinct *cis*-genes.

Diverging from mQTL studies that focus on variants with significant associations with CpG methylation, our study utilized variable

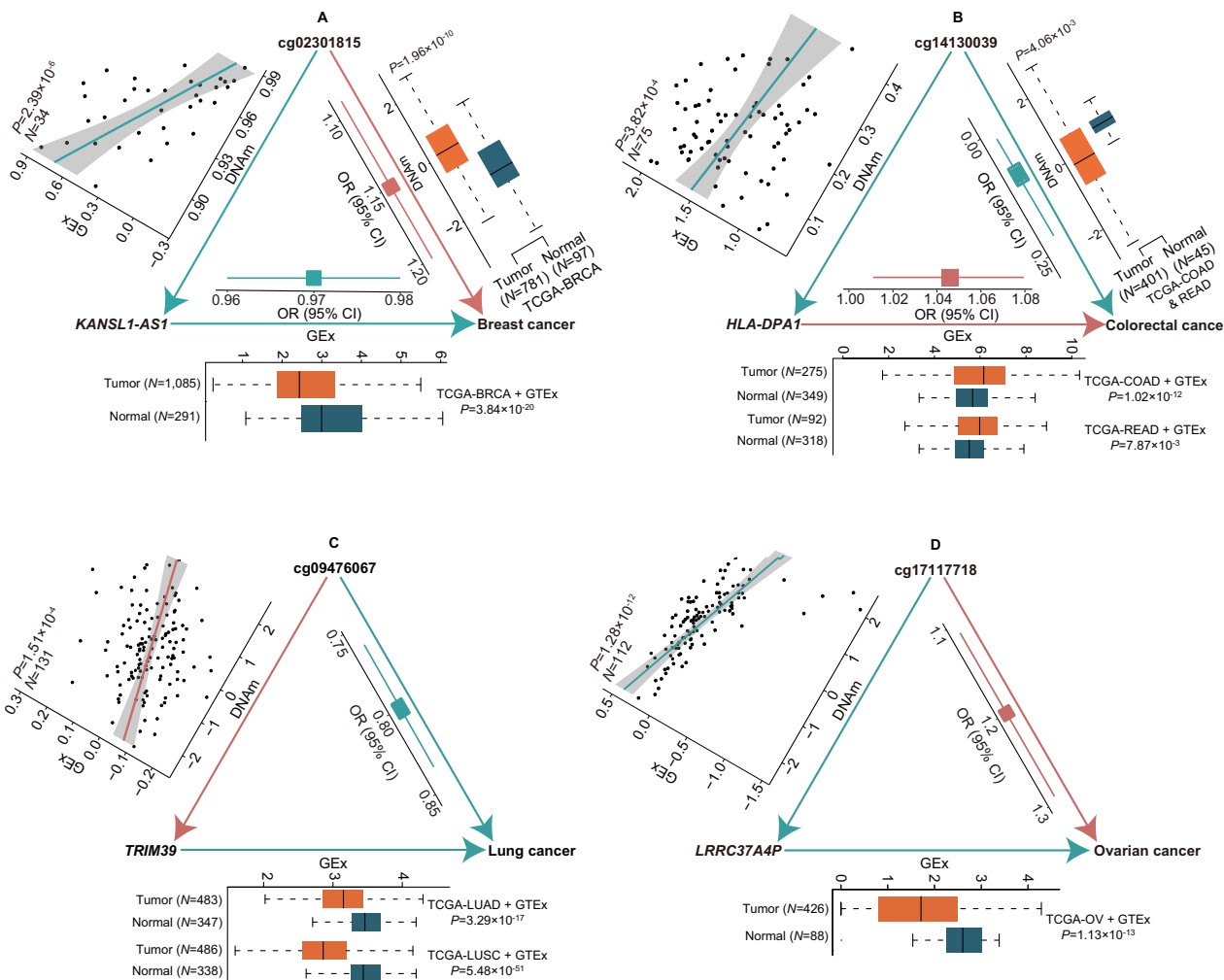

**Fig. 3 | Examples of CpG-gene-cancer trios suggesting DNA methylation influencing cancer risk by modulating *cis*-gene expression.** Association analyses of genetically predicted DNA methylation (DNAm) or gene expression (GEx) with cancer risk were performed using SPrediXcan. Differential DNAm or GEx analyses were conducted using linear mixed models. Association analyses between DNAm and GEx were performed using linear regression. Red arrows, lines, and blocks denote positive associations, while green ones denote negative associations. All statistical tests were two-sided and multiple comparisons were Bonferroni- or false discovery rate (FDR)-adjusted. The odds ratio (OR) and 95% confidence interval (CI) for cancer risk per standard deviation (SD) increase in genetically predicted DNAm or GEx are displayed as a block with error bands. In boxplots, boxes represent the interquartile range, black bars are medians, and whiskers extend at most 1.5 times the interquartile range. In the scatter plot displaying the association between DNAm and GEx, directly measured DNAm and GEx values after quantile- and inverse-normalization are presented. *N* number of samples, TCGA The Cancer Genome Atlas, GTEx Gene-Tissue Expression consortium, BRCA breast invasive carcinoma, COAD colon adenocarcinoma, READ rectum adenocarcinoma, LUAD lung adenocarcinoma, LUSC lung squamous cell carcinoma, OV ovarian serous cystadenocarcinoma. **A** DNAm at cg02301815 may elevate breast cancer risk by suppressing the expression of *KANSL1-AS1*. The sample size was 878 for tumor-normal differential DNAm analyses, 34 for DNAm-GEx correlation analyses, 1376 for tumor-normal differential GEx analyses, and 539,198 for association analyses of predicted DNAm and GEx with breast cancer risk, respectively. **B** DNAm at cg14130039 may decrease colorectal cancer risk by suppressing the expression of *HLA-DPA1*. The sample size was 446 for tumor-normal differential DNAm analyses, 75 for DNAm-GEx correlation analyses, 624 (TCGA-COAD + GTEx) and 410 (TCGA-READ + GTEx) for tumor-normal differential GEx analyses, and 254,791 for association analyses of predicted DNAm and GEx with colorectal cancer risk, respectively. **C** DNAm at cg09476067 may decrease lung cancer risk by promoting the expression of *TRIM39*. The sample size was 131 for DNAm-GEx correlation analyses, 830 (TCGA-LUAD + GTEx) and 824 (TCGA-LUSC + GTEx) for tumor-normal differential GEx analyses, and 887,170 for association analyses of predicted DNAm and GEx with lung cancer risk, respectively. **D** DNAm at cg17117718 may increase ovarian cancer risk by suppressing the expression of *LRCC37A4P*. The sample size was 112 for DNAm-GEx correlation analyses, 514 (TCGA-OV + GTEx) for tumor-normal differential GEx analyses, and 70,668 for association analyses of predicted DNAm and GEx with ovarian cancer risk, respectively. Differential DNAm analyses were unable to be performed for ovarian cancer-associated CpG because of the small sample size (*n* < 10) of the TCGA-OV DNA methylation datasets. Source data are provided as a Source data file.

selection algorithms that include all variants with any contributions to CpG methylation in prediction models, regardless of the statistical significance of their individual associations with CpG methylation. These algorithms, widely used in TWAS, have developed prediction models for a large number of genes, many of which lacked eQTLs[20,29]. In line with this, the number of CpGs with reliable prediction models established by the present study is 1.7 times greater than the number

of CpGs for which mQTLs were identified in the previous study using the same datasets[15]. Moreover, of these 478,360 CpGs for which our study built reliable models, only ~33% met the reliability criteria of $R > 0.10$ and two-sided $P < 0.05$ when predicted by the single best mQTLs. Furthermore, for these CpGs, our models showed significantly higher predictive accuracy compared to those based on the single best mQTLs. This is also consistent with TWAS findings that models

integrating multiple *cis*-variants outperformed those relying on the single best eQTLs[20,29]. Our results showcase the pronounced sensitivity and effectiveness of our method in capturing the genetic influences on DNA methylation than the single best mQTL approach.

Our preceding studies using blood-based models identified multiple CpGs associated with the risk of four cancers[16–19]. In the present study based on tissue-specific models, we identified almost 1.1, 1.4, 2.8, and 2.9 times more CpGs for breast, lung, ovarian, and prostate cancer, respectively, compared to our previous studies. Of the 3494 significant associations identified for these four cancers in the present study, more than 3192 (91.3%) were not found in our previous studies. This disparity might be attributed to the absence of 1311 CpGs (41.7% of 3192) in the Illumina 450 K array and/or the tissue-specific nature of these CpG. Importantly, of the remaining 302 CpGs that also showed in our previous studies, 283 (93.7%) exhibited consistent associations with the risk corresponding cancers at two-sided $P < 0.05$. This result suggests shared genetic effects on DNA methylation variations at these CpGs between tissues where cancers originate and blood, lending a support of the utility of these CpGs as non-invasive cancer biomarkers.

A recent study investigated genetically predicted colorectal tissue DNA methylation and colorectal cancer risk[4]. Our study of colorectal cancer utilized the same GWAS dataset, yet with improvements in prediction model development. First, we exclusively employed data of transverse colon tissues from cancer-free individuals, while the recent study involved various colorectal tissue types, including those adjacent to tumors of colorectal cancer patients. In addition, our study benefited from the inclusion of data from eight other distinct tissue types, enabling the development of colon-specific models while leveraging information from other tissues. In contrast, the recent study was confined to building single-tissue models. Finally, the current study, even with a more stringent threshold to select models ($R > 0.10$ and two-sided $P < 0.05$), established models for almost 6.5 times more CpGs (197,947 vs. 30,385) than the recent study. As a result, our association analyses identified nearly 1.5 times as many colorectal cancer-associated CpGs (792 vs. 501) and replicated almost 103 (20.5%) of the CpGs reported by the recent study[4].

Our study approach demonstrated high effectiveness in elucidating GWAS-identified cancer susceptibility loci, revealing association signals in a large number of known GWAS loci, especially those lacking TWAS hits. Despite employing nearly half the number of samples for prediction model development compared to gene- and splicing-based TWAS, the present study detected association signals in 1.2 times more GWAS loci (453 vs. 377). Notably, for prostate cancer, our study used 105 samples to develop prediction models and identified significant associations in 182 known GWAS loci, while TWAS, with 1.7 times more samples ($n = 180$), found signals in only 121 known loci. In addition, it is important to emphasize that nearly 32% of known GWAS loci containing cancer-associated CpGs were not reported by previous TWAS to have any genes or splicing variants associated with cancer risk.

While the vast majority of cancer-associated CpGs (94.4%) identified in our study are within known GWAS loci and their associations with cancer risk were likely driven by nearby GWAS signals, we still identified 92 CpGs in 55 loci situated >1 Mb away from any GWAS-identified risk variants. This result underlines the strength of our CpG-based association test approach in unveiling putative novel cancer susceptibility loci, expanding the reach of variant-based association studies in conventional GWAS. Moreover, more than 57% and 41% of these 92 CpG-cancer pairs showed a moderate to high possibility of colocalization, implying the potential of these CpGs as causal DNA methylation biomarkers for cancer risk and the plausible importance of these loci in cancer susceptibility. For instance, we found that DNA methylation at cg05649751 in the *12q24.11* locus was significantly associated (two-sided $P = 2.41 \times 10^{-7}$) and strongly colocalized (PP.H4 = 0.95) with increased prostate cancer risk. In addition, we detected a nominally significant positive correlation between cg05649751 methylation and the expression of one of its *cis*-genes, *SH2B3*, which encodes a critical negative regulatory protein in cytokine signaling and was reported to be associated with the risk of lung, colorectal, and breast cancer[32]. These results collectively suggest the potential involvement cg05649751 and *SH2B3* in the susceptibility to prostate cancer at *12q24.11*.

For 425 (~9.5%) of the 4461 cancer-associated CpGs, we pinpointed potential target genes whose expression significantly associated with DNA methylation of these CpGs in corresponding tissues. This observation is in alignment with previous findings that a large proportion of mQTL-GWAS colocalizations lack eQTL involvement[15]. These results suggest the potential presence of mechanisms that are alternative to gene expression regulation underlying variant-cancer associations identified by GWAS. Noteworthy, for 33 of these genes, we found evidence from CRISPR-Cas9 screening data supporting their essential roles in the proliferation of corresponding cancer cells. Further, 205 of these target genes were involved in 854 CpG-gene-cancer trios, implying the impacts of 309 CpG on cancer risk by regulating expression of these genes. Among them, 18 genes were essential for cancer cell proliferation, such as *NSF* for breast cancer, *CCND1* for renal cell cancer, *DHX16* for lung cancer, *CDC27* for ovarian cancer, and *CTDP1* for prostate cancer. Altogether, these findings demonstrated the capability of our study to identify functional genes that might be involved in putative genetic variants-DNA methylation-gene expression-cancer pathways.

Our study has notable strengths. First, we utilized Illumina MethylationEPIC BeadChip DNA methylation data from normal tissue samples of cancer-free individuals, enabling unbiased estimation of genetic determinants of DNA methylation at over 750,000 CpGs. In addition, both single- and joint-tissue prediction models were developed, considering both tissue-specific and shared genetic influences on DNA methylation to improve prediction accuracy. Moreover, despite a smaller sample size involved in model development, our study identified significant associations in a larger number of known GWAS loci compared to gene- and splicing-based TWAS. Further, the replication of a substantial proportion of associations using external data of tumor and adjacent normal tissue samples strengthened the validity of our findings. Finally, the discovery of CpG-gene-cancer trios provided mechanistic insights into the critical roles of epigenetics in the genetic etiology of cancer.

Several limitations should be noted. First, the sample size for prediction model development, despite being one of the largest for many tissue types, remained relatively small compared to the extensive number of examined CpGs. Expanding the sample sizes, particularly for tissues with limited samples, would enhance model precision and possibly unveil additional significant associations. Second, although colocalization analyses provided evidence supporting the existence of shared causal variants between CpG methylation and cancer risk for over 32% of significant CpG-cancer association we identified, the potential causal relationship between these CpGs and cancer risk could not be fully elucidated. Finally, the differential methylation analyses using TCGA data were limited by the small sample size and the potential differences in DNA methylation profiles between normal tissues adjacent to tumors and those obtained from cancer-free subjects. Future studies employing normal tissue samples from cancer-free individuals, coupled with functional experiments, are needed to further corroborate our findings.

In summary, we identified >4400 CpGs showing tissue-specific associations with cancer risk, nearly 300 of which may influence cancer risk by regulating neighbor gene expression. Our findings emphasize the effectiveness of multi-omics integration in cancer biomarker discovery and enhance our comprehension of the critical role of genetics and epigenetics in cancer etiology.

## Methods

### Statistics and reproducibility

All analyses in this study utilized publicly available data, and therefore, no statistical method was used to predetermine sample size. The development of DNA methylation prediction models requires subjects with both DNA methylation and genetic data, leading to the exclusion of 57 subjects lacking genetic data, as detailed in "Data Acquisition". The experiments were not randomized. The investigators were not blinded to allocation during the experiments and outcome assessment, because the data are not from controlled randomized studies. Association analyses of genetically predicted DNA methylation or gene expression with cancer risk were conducted using SPrediXcan[23]. Differential DNA methylation or gene expression analyses utilized linear mixed models, while association analyses between DNA methylation and gene expression were performed using linear regression. All statistical tests were two-sided, and multiple comparisons were Bonferroni- or FDR-adjusted. Self-reported sex was considered in all analyses. Analyses stratified by sex were not performed except for sex-specific cancer types due to the unavailability of sex-stratified cancer GWAS data. Python (v3.6.3) and/or R (v3.6.0) were used for all analyses. All codes and data necessary to reproduce the study findings are publicly available on Zenodo[33]. Further details are available in the corresponding "Methods" section.

### Data acquisition

All data used in the present study are publicly available[4,6,8,10,15,26,27,29,31,33-38]. Whole-genome sequencing (WGS) data of blood samples and Illumina MethylationEPIC BeadChip DNA methylation data of normal tissue samples from GTEx (v8) were used as references to build DNA methylation prediction models[10,15]. Genotype and phenotype data were downloaded from dbGaP (phs000424.v8.p2). Normalized DNA methylation data of nine tissue types, including breast, colon, kidney, lung, ovary, prostate, testis, whole blood, and muscle, was obtained from GEO (GSE213478). Detailed information on sample preparation, sequencing, and data processing were described elsewhere[10,15]. Briefly, WGS libraries built from blood DNA samples from 424 donors were sequenced on the Illumina HiSeq X or HiSeq 2000 platform at the Broad Institute with a median coverage of ~32X. Genotype data was extracted and non-palindromic variants with missing data < 5%, minor allele frequency (MAF) > 5%, and Hardy-Weinberg equilibrium (HWE) $P > 10^{-4}$ were retained for subsequent analyses. DNA methylation profiling was performed using the Illumina MethylationEPIC BeadChip based on 987 tissue DNA samples across nine unique tissue types obtained from these 424 subjects. The R package *ChAMP* (v2.8.6)[39] was utilized to process raw data to exclude low-quality samples and CpGs and estimate DNA methylation $\beta$ values[15]. After background correction using the single sample normal-exponential out-of-band (ssnoob) method implemented in the R package *minfi* (v1.36.0)[40], $\beta$ values were normalized using the BMIQ method[15]. We further removed 57 subjects lacking genetic data, along with DNA methylation data of the 131 samples donated by them. The downstream analyses included DNA methylation data of 754,119 CpGs among 856 samples from 367 subjects aged 20–70 years old (207 males and 160 females based on self-reported sex), along with paired genotype data of ~5.1 million variants (IQR: 4.6-5.6). These subjects were of diverse ancestries, including European ($n = 318$), African ($n = 44$), Asian ($n = 4$), and American Indian/Alaska Native ($n = 1$).

Summary statistics of GWAS for breast, colorectal, renal cell, lung, ovarian, prostate, and testicular germ cell cancers were acquired from different sources[2–8] (Fig. 1B and Supplementary Data 1). Except for renal cell and testicular germ cell cancers, GWAS data from at least one non-European population were available for the other five cancer types. For each of these five cancers, data from different ancestral populations were combined by fixed-effects

meta-analyses using METAL[41]. Finally, data of breast cancer included 158,742 cases and 380,456 controls, composed of 133,511 cases and 291,090 controls of European ancestry from the Breast Cancer Association Consortium (BCAC)[26,42] and UK Biobank[26,35], 4832 cases and 3020 controls of African ancestry from BCAC[42], and 20,399 cases and 86,346 controls of Asian ancestry from BCAC[42] and Biobank Japan[37]. Data of colorectal cancer included 100,204 cases and 154,587 controls from 31 studies[4], comprising 78,473 cases and 107,143 controls of European ancestry from 17 studies, and 21,731 and 47,444 of Asian Ancestry from 14 studies. Data of renal cell cancer included 10,784 cases and 20,406 controls of European ancestry from six studies[8]. Data of lung cancer included 50,503 cases and 836,667 controls, composed of 38,422 cases and 677,930 controls of European ancestry from the Transdisciplinary Research of Cancer in Lung of the International Lung Cancer Consortium (TRICL-ILCCO)[34], the Lung Cancer Cohort Consortium (LC3)[34], UK Biobank[35], and FinnGen (R9)[36], and 12,081 cases and 158,737 controls of Asian ancestry from the Nanjing Medical University (NJMU)[43], the Female Lung Cancer Consortium in Asia (FLCCA)[44], and Biobank Japan[37]. Data of ovarian cancer included 25,644 cases and 45,024 controls from the Ovarian Cancer Association Consortium (OCAC), comprised of 22,406 cases and 40,941 controls of European ancestry[6] and 3238 cases and 4083 controls of Asian ancestry[38]. Data of prostate cancer included 156,319 cases and 788,443 controls from 151 studies[9], composed of 122,188 cases and 604,640 controls of European ancestry from 95 studies, 19,391 cases and 61,608 controls of African ancestry from 42 studies, 10,809 cases and 95,790 controls of Asian ancestry from six studies, and 3931 cases and 26,405 controls of Hispanic or Latino ancestry from eight studies[9]. Data of testicular germ cell cancer included 10,156 cases and 17,979 of European ancestry from The Testicular Cancer Consortium (TECAC)[7].

### DNA methylation prediction model development

For each tissue, BMIQ-normalized DNA methylation $\beta$ values were inverse-normalized within each CpG and regressed on covariates to get residuals. These covariates included top five genetic principle components (PCs), Probabilistic Estimation of Expression Residuals (PEER)[45] factors ($n = 5$ for breast, kidney, testis, muscle, and whole blood; $n = 20$ for colon, lung, ovary, and prostate), self-reported sex (only for colon, kidney, lung, muscle, whole blood), indicators for WGS sequencing platform (HiSeq X or HiSeq 2000), and library construction protocol indicator (PCR based or PCR-free). For each CpG, we constructed two prediction models using different strategies, including the elastic net method ($\alpha = 0.50$) exclusively based on data from the specific tissue type, and the Unified Test for MOlecular SignaTures (UTMOST [2023 release])[21] method capturing genetic effects on DNA methylation shared across various tissues. It's important to note that UTMOST models are also fundamentally tissue-specific because the information from other tissues could be incorporated only when there is a certain degree of similarity in genetic effects on DNA methylation with the primary tissue in focus. To evaluate the predictive performance of our method, which uses multiple *cis*-variants, against the approach relying solely on the single best *cis*-mQTL, we established an additional model for each CpG using its best *cis*-mQTL. For all three strategies, genetic variants within the 500Kb flanking region of each CpG were utilized to select those most predictive variant(s) for predicting the inverse-normalized DNA methylation residuals with five-fold cross-validation. Models with $R > 0.10$, denoting 10% positive correlation between genetically predicted and measured DNA methylation levels, and two-sided $P < 0.05$ were considered reliable[21]. CpGs for which at least one model, either developed by elastic net or UTMOST, met these criteria were included in downstream association analyses. When both models are qualified, only the one with the higher R value was used for these analyses.

## Association analyses between genetically predicted DNA methylation and cancer risk

SPrediXcan[23] was utilized to assess associations between genetically predicted DNA methylation level at CpGs and cancer risk. The association Z score was calculated following the below formula, in which $W_{S-m}$ represents the weight of variant $S$ on DNA methylation levels at CpG $m$, $\hat{\sigma}_S$ and $\hat{\sigma}_m$ represents estimated variances of variant $S$ and CpG $m$, and $\hat{\beta}_S$ and $se(\hat{\beta}_S)$ represents effect size and standard error of the association between variant $S$ and cancer risk, respectively.

$$Z_m = \sum_{s \in Model_m} W_{S-m} \frac{\hat{\sigma}_S}{\hat{\sigma}_m} \frac{\hat{\beta}_S}{se(\hat{\beta}_S)} \quad (1)$$

Association analyses and Bonferroni-correction were conducted for each cancer type separately and significant associations were identified at Bonferroni-corrected two-sided $P < 0.05$, corresponding to $4.93 \times 10^{-7}$ (0.05/101,497) for breast cancer, $2.53 \times 10^{-7}$ (0.05/197,947) for colorectal cancer, $3.98 \times 10^{-7}$ (0.05/125,745) for renal cell cancer, $2.55 \times 10^{-7}$ (0.05/195,764) for lung cancer, $2.66 \times 10^{-7}$ (0.05/187,911) for ovarian cancer, $3.28 \times 10^{-7}$ (0.05/152,341) for prostate cancer, and $4.22 \times 10^{-7}$ (0.05/118,568) for testicular germ cell cancer.

For each significant CpG-cancer association identified, we performed colocalization analyses implemented in the R package *Coloc* (v5.2.3)[24] to ascertain whether CpG methylation and cancer risk might be affected by a causal variant or by different variants in LD within the CpG's *cis*-region. A moderate probability of colocalization was indicated by PP.H4 > 0.50[15], and a high probability by PP.H4 > 0.80. We then evaluated the independence of identified CpG-cancer associations from their nearest GWAS signals. For each CpG-cancer pair, we first conducted stepwise model selection[25] to identify variants that were independently associated with cancer risk at $P < 5.00 \times 10^{-8}$ in the CpG's nearest cancer susceptibility locus identified by GWAS. Then for each variant included in the CpG's prediction model, its association with cancer risk conditioning on the variants identified in the first step was evaluated using GCTA-COJO[25]. Finally, SPrediXcan analysis was conducted using the summary statistics generated in the second step and the Bonferroni-corrected thresholds used in our primary analyses were applied to determine significance.

For CpGs significantly associated with cancer risk, we assessed their differential DNA methylation between tumor and adjacent normal tissues using Illumina HumanMethylation450 BeadChip DNA methylation data from TCGA[27]. Such analyses were not conducted for ovarian and testicular germ cell cancers due to the limited number of available samples ($n < 10$) with DNA methylation data in TCGA. DNA methylation $\beta$ values of 485,577 CpGs and patient information were obtained from the National Cancer Institute (NCI) Genomic Data Commons Data Portal. For each cancer, data from subtypes were combined to improve statistical power. Specifically, we combined TCGA-COAD (colon adenocarcinoma) and TCGA-READ (rectum adenocarcinoma) for colorectal cancer, TCGA-KIRC (kidney renal clear cell carcinoma) and TCGA-KIRP (kidney renal papillary cell carcinoma) for renal cell cancer, and TCGA-LUAD (lung adenocarcinoma) and TCGA-LUSC (lung squamous cell carcinoma) for lung cancer. DNA methylation $\beta$ values were quantile-normalized between samples. Then for each CpG site, we conducted inverse-normalization on the DNA methylation values of all samples. For each CpG, differential DNA methylation analysis was performed by fitting a linear mixed-effects model with tissue status (tumor/adjacent normal) modeled as a random effect, adjusting for age, race, sex (where applicable), sample type indicator (Formalin-Fixed Paraffin-Embedded or not), and top three DNA methylation PCs[46]. For analyses of colorectal, renal cell, and lung cancers, cancer subtype and self-reported sex were additionally adjusted. For lung cancer, smoking status (current/former/never) and pack-year of smoking were further adjusted. Numbers of tumor and adjacent normal tissue samples included in the final analyses were 781 vs. 97 for breast, 404 vs. 45 for colorectal, 597 vs. 205 for renal cell, 839 vs. 74 for lung, and 502 vs. 50 for prostate cancer. CpGs showing a differential DNA methylation at two-sided $P < 0.05$ with directions of effect sizes consistent with Z scores of their associations with cancer risk in our primary analyses were considered validated.

## Identifying CpG-gene-cancer Trios

For each cancer-associated CpG, eQTM analyses were conducted to search for potential target genes in its 500Kb flanking region using data of the corresponding tissue. Directly measured DNA methylation data, which were utilized for residual calculation during prediction model development for cancer risk analyses, along with gene expression data downloaded from GTExPortal, were employed for these analyses. In total, 34 breast, 75 colon, 131 lung, 112 ovary, 44 prostate, and 25 testis tissue samples with paired DNA methylation and gene expression data available were involved in this analysis. For each tissue type, genes with ≥6 read and >0.10 Transcript Per Million (TPM) were retained and TPM values were quantile-normalized between samples. Then for each gene, we performed inverse-normalization on expression values of all samples. Finally, for each CpG-gene pair, a linear regression model was fitted with inverse-normalized DNA methylation values as the exposure and inverse-normalized gene expression values as the outcome. Five DNA methylation PEERs and five gene expression PEERs were additionally adjusted. Finally, FDR correction was applied to the nominal *P*-values and significant associations were identified at FDR-corrected two-sided $P < 0.05$. Such analyses could not be performed for kidney tissue due to the extremely small sample size ($n = 5$). To address this, results of CpG-gene associations based on data of 414 non-neoplastic kidney tissue samples were accessed from a previous study[47].

For genes significantly associated with cancer-associated CpGs, we first evaluated their effects on essentiality for proliferation of corresponding cancer cells using CRISPR-Cas9 screening data. CERES values of these genes in cells relevant to breast ($n = 48$), colon and rectum ($n = 57$), kidney ($n = 32$), lung ($n = 114$), ovary ($n = 57$), and prostate ($n = 10$) were obtained from the DepMap Public 23Q2 release. For a particular cancer, genes with a median CERES value < -0.50 across all cell lines were considered essential for proliferation[28]. Next, these genes were investigated for their genetically predicted expression in association with cancer risk. Single- and cross-tissue gene expression prediction models developed using GTEx (v8) data via elastic net and Joint-Tissue Imputation (JTI) approaches were acquired from PredictDB[29] and Zenodo[30], respectively. For each of the genes that had at least one model with $R^2 > 0.01$ and two-sided $P < 0.05$, only the model with higher R² value was used in association analyses with cancer risk using SPrediXcan[23]. Further, we examined the differential expression of these genes between tumor and normal tissues. Results from the Gene Expression Profiling Interactive Analysis (GEPIA2) web server[31] were used. Data curation and analyses by the GEPIA2 team are described in detail elsewhere[31]. Briefly, for each of 33 cancer types, raw RNA-seq data from TCGA and GTEx were processed using a uniform pipeline and differential expression analyses between TCGA tumor tissues and TCGA adjacent normal tissues plus GTEx normal tissues were conducted using the R package *limma* (v3.18)[31]. For both analyses, FDR-correction was applied for each cancer separately and FDR-corrected two-sided $P < 0.05$ was used to determine statistical significance.

Finally, to identify CpG-gene-cancer trios supporting DNA methylation of cancer-associated CpGs influencing cancer risk by modulating neighboring gene expression, we jointly analyzed results from CpG-cancer, CpG-gene, and gene-cancer associations. Specifically, a valid CpG-gene-cancer trio requires the directions of CpG-cancer, CpG-gene, and gene-cancer associations to be reasonably consistent. Therefore, the gene-cancer association direction is

determined by the combination of the directions of CpG-cancer and the CpG-gene associations. We evaluated gene-cancer associations through two methods: assessing genetically predicted gene expression in association with cancer risk and comparing gene expression between cancer and normal tissues. For a trio to be established, the association direction from either of these methods must correspond with the direction indicated by the CpG-cancer and CpG-gene associations.

## Reporting summary
Further information on research design is available in the Nature Portfolio Reporting Summary linked to this article.

## Data availability
The publicly available genotype, DNA methylation, and gene expression data of GTEx participants used in this study are available in the dbGaP and GEO under accession code phs000424.v8.p2[10,15] and GSE213478[10,15], respectively. The publicly available DNA methylation data of TCGA participants used in this study are available in the NCI Genomic Data Commons Data Portal [https://portal.gdc.cancer.gov/][27]. The publicly available GTEx v8-based gene expression and splicing prediction models used in this study are available in PredictDB [https://predictdb.org/][29]. The publicly available differential gene expression data used in this study are available in GEPIA2[31]. The publicly available summary statistics of cancer GWAS used in this study are available in Zenodo (accession code 7814694) for breast cancer[26], GWAS catalog (accession code GCST90129505) for colorectal cancer[4], dbGaP (accession code phs001736.v2.p1) for renal cell cancer[8], GWAS catalog (accession code GCST004746[34, 48]), UK Biobank data from the Neale lab [https://www.nealelab.is/uk-biobank][35], the FinnGen website [https://r9.finngen.fi/][36], and the Biobank Japan website [https://pheweb.jp/][37] for lung cancer, OCAC website [https://ocac.ccge.medschl.cam.ac.uk/data-projects/] for ovarian cancer[6,38], GWAS catalog (accession code GCST90274713) for prostate cancer[38], and dbGaP (accession code phs001349) for testicular germ cell cancer[38]. The DNA methylation prediction models generated in this study have been deposited in Zenodo (accession code 10810820)[33]. The remaining data are available within the Article, Supplementary Information or Source Data files. Source data are provided with this paper.

## Code availability
The codes that could be used to replicate our findings have been deposited in Zenodo under accession code 10810820[33].

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

## Acknowledgements

Y.Y. is partially supported by the NIH grant R00CA248822. This work was also supported in part by NIH grants R01CA249863 (MPIs: C.Q. and L.J.) and R01CA247987 (MPIs: L.J. and Y.F.). The funders had no role in study design, data collection and analysis, decision to publish, or preparation of the manuscript. Data analyses were conducted on the Rivanna High-Performance Computing (HPC) system at University of Virginia and the Advanced Computing Center for Research and Education (ACCRE) HPC system at Vanderbilt University.

## Author contributions

Conception and design of the study: Yaohua Yang, Qiuyin Cai, and Jirong Long. Data analyses: Yaohua Yang and Yaxin Chen. Interpretation of findings: Yaohua Yang, Yaxin Chen, Qiuyin Cai, and Jirong Long. Manuscript writing: Yaohua Yang. Substantive manuscript revision: Yaohua Yang, Yaxin Chen, Shuai Xu, Xingyi Guo, Guochong Jia, Jie Ping, Xiang Shu, Tianying Zhao, Fangcheng Yuan, Gang Wang, Yufang Xie, Hang Ci, Hongmo Liu, Yawen Qi, Yongjun Liu, Dan Liu, Weimin Li, Fei Ye, Xiao-Ou Shu, Wei Zheng, Li Li, Qiuyin Cai, and Jirong Long. Overall supervision of the project: Yaohua Yang.

## Competing interests

The authors declare no competing interests.
