## [Peer Review File · Nature Communications]

Integrating multi-omics data to identify tissue-specific DNA methylation biomarkers for cancer riskReviewers' Comments:

Reviewer #1:

Remarks to the Author:

Overall comments:

The paper "Integrating multi-omics data to identify tissue-specific DNA methylation biomarkers for cancer risk" tests the association of genetically predicted CpG methylation with cancer risk in 7 tissues (breast, ovary, lung, colorectal, prostate, kidney, testis). They use the GTEx data to build ~6 million prediction models, one for each CpG on the MethylationEPIC array for each tissue type. Their comprehensive analysis identifies 35 putative novel loci with CpGs predicting cancer risk. Strengths include the use of normal tissue from non-cancer patients for building the genetic risk prediction models, and use of a model that fits all tissues simultaneously. A weakness (albeit a weakness of the field in general) is that the models are only built and applied to GWAS from non-European descendants, colon being an exception. The paper is well-organized and well-written.

General comments:

It is disturbing to read the results from a study that excludes samples from non-European descent as if they were unnecessary, and selects only GWAS from individuals with European ancestry. This needs explaining in the manuscript. Please describe the choice of your population and reasons for the selection of samples analyzed in this study in light of the new guidelines on population descriptors in genetics and genomics research published by the National Academies of Sciences, Engineering & Medicine (<https://nap.nationalacademies.org/catalog/26902/using-population-descriptors-in-genetics-and-genomics-research-a-new>). Some guidance offered in Chapter 5 for genomic study type 4, prediction for complex and polygenic traits, seems directly relevant.

Specific Comments:

1. Introduction, line 48. Please provide the sample size for the study using the MethylationEPIC array.
2. Lines 121-125 Please re-write sentence. How do you get the summary > 45% of known GWAS loci?
3. Line 161, Move '(Table S8-S12)' from line 163 one sentence earlier to line 161.
4. Lines 215-226. This paragraph gives me no understanding of why your results are better and more correct than reference 14. Please report the number of overlapping samples analyzed in both papers and a reason for why your modeling approach is more sensitive.
5. Line 233-234, Please give the number of CpGs not on the 450K array.
6. Line 254 is unclear.
7. Line 260, known loci from what resource? This study or others published before?
8. Line 264, Do you mean cancer-associated CpGs identified in this study? Please be specific about the origin of this statistic.
9. Line 278 remove the word "most". Your study is not designed to answer the question about most variant-cancer associations.
10. Line 282, what target genes? How do you get here from ~12% of CpGs? Maybe you should give the number of CpGs before the ~12%
11. Data acquisition text is problematic for genomic descriptors. How do you drop from 1000 samples on 424 subjects to 738 samples on 317 samples is omitting a lot of samples and individuals
12. Line 345. Please create a Supplementary table with the information on cancer type/subtypes, study design with sample sizes and reference. Please refer to this in the paper along with Fig 1b.
13. Line 443-444, it seems worth mentioning what is different in this analysis so that you now include non-European descent subjects whereas before they were omitted.
14. Line 480-482 this sentence is not clear.
15. Figure 3D Is there any TCGA data for the boxplots?
16. Supplemental Table 1 should be provided as a spreadsheet so that the columns are not cutoff due to the page size.
17. Supplemental Table S9. Define tumor subtype. If it's colon and rectal, the word 'sites' is more appropriate than 'subtypes'.

18. Table S11 footnote, typo. Why use meta-analysis here when earlier tables used subtype adjustment?

Typos:

Line 82, no s on models

Line 86, issue should be tissue

Line 100 except not expect, and fix percentages.

Line 105, exclusive not exclusively,

Line 109, cancer not caner

Line 127, the capability of the approach

Line 159, MethylationEPIC and HumanMethylation450 spelling

Reviewer #2:

Remarks to the Author:

The authors developed models to pinpoint genetically influenced CpGs, utilizing paired methylation arrays and whole-genome sequencing (WGS) data from seven diverse tissues in the GTEx dataset. Subsequently, they utilized cancer Genome-Wide Association Study (GWAS) summary statistics specific to each tissue to identify CpGs associated with cancer risk. Within 35 potential new loci, 57 CpGs were discovered, showing significant associations with cancer susceptibility through nearby genes.

Examining the relationship between genotype and DNA methylation is a crucial area of study that can bridge the gap between genotype, phenotype, and diseases. These analyses hold promise for enhancing our understanding of cancer risk by exploring the interplay between genetics and epigenetics. However, the manuscript fails to clarify the conceptual distinctions between genetically influenced CpGs and mQTLs. Additionally, it remains unclear how much this method improves upon canonical QTL-based methods. The lack of adequate benchmarking undermines the credibility of the newly identified disease-relevant CpGs. Further comments are outlined below:

The term 'genetically determined methylation' used in this study is inappropriate and open to debate, as the study examines general associations rather than causality for the identified CpG sites.

These genetically associated CpGs were pinpointed using both tissue-specific and whole tissue models. The authors assert that the whole tissue model can leverage information from other tissues to identify tissue specific CpGs, but this reasoning needs explicit clarification.

Relevance exploration for genetically determined CpGs involved utilizing summary statistics of susceptibility from different cancers. Although GWAS studies from European ancestry were employed to mitigate differences in genetic architecture, specific effects related to linkage disequilibrium persisted, and determining the likely causal SNPs remained challenging.

Moreover, the analysis and result interpretation are challenging to follow because the authors have structured the paper assuming a high level of familiarity with their studies, making it difficult for readers unfamiliar with their work to grasp the content. It would be helpful, for instance, to provide a brief introduction to the software and methods used before delving into result interpretation and presentation. The term "predicted expression" (mentioned in line 183) lacks clarity. Also, the predicted value appeared multiple times and referring to various aspects without clear definition.

Minors:

The footnotes accompanying the tables and the tables within the PDF file are not comprehensive, leaving important information out.

the explanations regarding long-range associations (those beyond 1Mb) are insufficient. The figure legends lack crucial details. For instance, in Figure 1 (lines 655-659), it is unclear what the range of values represents in panel A, and the meaning of 'n' in panel B is not explained.

Authors' Responses to Comments from Reviewers

Responses to Reviewer #1.

Overall comments: *The paper “Integrating multi-omics data to identify tissue-specific DNA methylation biomarkers for cancer risk” tests the association of genetically predicted CpG methylation with cancer risk in 7 tissues (breast, ovary, lung, colorectal, prostate, kidney, testis). They use the GTEx data to build ~6 million prediction models, one for each CpG on the MethylationEPIC array for each tissue type. Their comprehensive analysis identifies 35 putative novel loci with CpGs predicting cancer risk. Strengths include the use of normal tissue from non-cancer patients for building the genetic risk prediction models, and use of a model that fits all tissues simultaneously. A weakness (albeit a weakness of the field in general) is that the models are only built and applied to GWAS from non-European descendants, colon being an exception. The paper is well-organized and well-written.*

Response: We recognize the limitation of predominantly using data from European descendants, which is a general issue in genetics research field. To address this concern, we have made substantial improvements to our study by including data from subjects of diverse ancestries and re-conducting all the analyses described in the original manuscript. Specifically, for the development of DNA methylation prediction models, we incorporated data from all GTEx subjects of all ancestries with both genetic and DNA methylation data. In addition, we updated the For Specifically, for colorectal and prostate cancers, we used to-date the largest cross-ancestry GWAS data from recently published studies, in which data for colorectal cancer included European- and Asian-ancestry subjects (PMID: 36539618), while that for prostate cancer include European, African, Asian, and Hispanic/Latino subjects (PMID: 37945903). For breast, ovarian, and lung cancers, we performed meta-analyses to combine data from different ancestral populations. For renal cell and testicular germ cell cancers, data of European descendants were used due to the unavailability of data from non-European populations. The revised manuscript (**Lines** 423-449) and **Supplementary Table S1** provide detailed information on these updated datasets. Using the newly generated prediction models and GWAS data, we re-conducted all analyses. We identified 4,461 significant associations, compared to the 2,776 in our original work, with the most substantial increase seen in prostate cancer, attributed to the almost doubled number of cases in GWAS. Notably, 2,184 (78.7%) of the 2,776 originally identified associations were also investigated in the current analyses, with 97.7% (2,133/2,184) showing $P < 0.05$ and 80.5% (1,795/2,184) showing Bonferroni-corrected $P < 0.05$. These efforts not only reinforce our initial findings but also enhance statistical power, leading to the discovery of a larger number of CpGs significantly associated with cancer risk. These improvements demonstrate our commitment to addressing ancestry representation in genetic studies, elevating the quality and impact of our research.

General comments: *It is disturbing to read the results from a study that excludes samples from non-European descent as if they were unnecessary, and selects only GWAS from individuals with European ancestry. This needs explaining in the manuscript. Please describe the choice of your population and reasons for the selection of samples analyzed in this study in light of the new guidelines on population descriptors in genetics and genomics research published by the National Academies of Sciences, Engineering & Medicine (<https://nap.nationalacademies.org/catalog/26902/using-population-descriptors-in-genetics-and-genomics-research-a-new>). Some guidance offered in Chapter 5 for genomic study type 4, prediction for complex and polygenic traits, seems directly relevant.*

Response: We thank the reviewer for highlighting the critical issue of ancestry representation in genetic studies. Our initial focus on European descent was influenced by the composition of GTEx subjects, over 90% of whom were of European ancestry, and the fact that GWAS for the seven cancers we investigated were mainly conducted in European populations. However, recognizing the vital importance of including a broader range of ancestries, as highlighted in the guidelines from the National Academies of Sciences, Engineering, and Medicine, we have expanded our analysis to include subjects from all ancestries present in the GTEx data and re-conducted all analyses outline in our original manuscript. This not only corroborated a substantial proportion of our initial findings but also led to the identification of additional CpGs significantly associated with cancer risk. These improvements, described in detail in our response

to the **Overall comment**, are reflected in the revised manuscript, ensuring that our research adheres to the latest standards for diversity and inclusivity in genetic research.

Specific comments 1: *Introduction, line 48. Please provide the sample size for the study using the MethylationEPIC array.*

Response: The study (PMID: 36510025) employed the Illumina MethylationEPIC BeadChip to profile DNA methylation in 987 samples from 424 subjects, representing nine distinct tissue types. We have added the number of samples and subjects in the revised manuscript (**Line 81**).

Specific comments 2: *Lines 121-125 Please re-write sentence. How do you get the summary > 45% of known GWAS loci?*

Response: To assess whether the cancer-associated CpGs identified in our study are situated in known GWAS loci, we manually curated all GWAS-reported cancer susceptibility variants based on GWAS catalog. We considered a cancer-associated CpG which is less than 1 mega base (Mb) way from its nearest cancer risk variant as potentially contributing to the genetic effects on cancer risk in the corresponding locus. This percentage was derived by dividing the number of loci with at least one cancer-associated CpG by the total number of GWAS-identified loci. In our revised manuscript, we identified 4,461 CpGs significantly associated with cancer risk. Among them, 4,210 CpGs (94.3%) are located within established cancer risk loci identified by GWAS. Among the 801 GWAS-identified cancer risk loci curated by us, we identified at least one CpG significantly associated with cancer risk in 453 loci (56.5%). We have made corresponding revisions in the manuscript (**Lines 167-168, 186**). In addition, we have included **Supplementary Tables S9-S15** to present the GWAS-identified cancer susceptibility loci and their leading variants, along with references and the representative cancer-associated CpG for each locus.

Specific comments 3: *Line 161, Move '(Table S8-S12)' from line 163 one sentence earlier to line 161.*

Response: Thanks! We have made this revision in the revised manuscript (**Line 218**).

Specific comments 4: *Lines 215-226. This paragraph gives me no understanding of why your results are better and more correct than reference 14. Please report the number of overlapping samples analyzed in both papers and a reason for why your modeling approach is more sensitive.*

Response: Unlike the conventional mQTL approach that only considers the effect of a single genetic variant on CpG methylation, our approach aggregates the effects of multiple *cis*-genetic variants to improve CpG methylation prediction. In our revised manuscript, we directly compared our approach with the mQTL study (reference 14, PMID: 36510025) by using the same dataset. We evaluated prediction performance of our prediction models against the single best mQTLs. Among the seven tissues investigated, our prediction method established reliable models ($R > 0.1$ and $P < 0.05$) for an average of 154,253 CpGs per tissue, outperforming the single best mQTL approach, which was only able to reliably predict 33.1% of these CpGs. Notably, our models consistently demonstrated significantly higher predictive accuracy (R values) than the single best mQTL SNPs across all tissues (all P values from paired t-test $< 2.20 \times 10^{-16}$). These results demonstrate the increased sensitivity and predictive accuracy of our prediction strategy compared to the mQTL approach. Detailed information on these analyses and results have been incorporated in **Results (Lines 133-141)**, **Discussion (Lines 281-295)**, **Methods (Lines 464-469)** of the revised manuscript and **Supplementary Figure S1**.

Specific comments 5: *Line 233-234, Please give the number of CpGs not on the 450K array.*

Response: Overall, 47.6% (413,894) of the 868,565 CpGs on the Illumina EPIC Array are not included in the 486,428 CpGs on the Illumina 450K Array. In our updated analysis, we identified 3,494 CpGs significantly associated with breast, ovarian, lung, and prostate cancers, 3,192 CpG of which were not found to be associated with these cancers by our previous studies using Illumina 450K-array-based blood DNA methylation data. Of these 3,192 CpGs, 1,311 (41.7%) are not on the Illumina 450K array. We have clarified this in the revised manuscript (**Lines 302-303**).

Specific comments 6: *Line 254 is unclear.*

Response: We have revised this sentence to reflect our findings more accurately: “Our study approach demonstrated high effectiveness in elucidating GWAS-identified cancer susceptibility loci, revealing association signals in a large

number of known GWAS loci, especially those lacking TWAS hits.” (Lines 324-326). We hope this revised sentence now effectively conveys our intended message.

Specific comments 7: *Line 260, known loci from what resource? This study or others published before?*

Response: Please see response to **Specific comments 2**.

Specific comments 8: *Line 264, Do you mean cancer-associated CpGs identified in this study? Please be specific about the origin of this statistic.*

Response: Yes, we mean cancer-associated CpGs identified in the present study and have clarified this in the revised manuscript (Line 335).

Specific comments 9: *Line 278 remove the word “most”. Your study is not designed to answer the question about most variant-cancer associations.*

Response: We have removed the word “most” from this sentence in the revised manuscript (Lines 356-357).

Specific comments 10: *Line 282, what target genes? How do you get here from ~12% of CpGs? Maybe you should give the number of CpGs before the ~12%.*

Response: We revised this section to clarify this. The term “potential target genes” refers to genes whose expression levels were significantly associated with the DNA methylation levels of cancer-associated CpGs in cancer-relevant tissues. We also added the number of CpGs before the percentage in the revised manuscript (Lines 352-354).

Specific comments 11: *Data acquisition text is problematic for genomic descriptors. How do you drop from 1000 samples on 424 subjects to 738 samples on 317 samples is omitting a lot of samples and individuals.*

Response: As we clarified in our response to the **Overall comments**, in our initial analyses, samples from non-European descendants or subjects without genetic data were excluded. In the revised manuscript starting from 987 tissue samples from 424 cancer-free GTEx subjects, we only excluded 131 samples from 57 subjects without genetic data and kept the remaining 856 samples from 367 subjects for DNA methylation prediction model development. These 367 subjects comprised of 318, 44, and four subjects of European, African, and Asian ancestry, and one subject of American Indian or Alaska Native ancestry. We clarified this in the **Results** (Lines 111-114) and **Methods** (Lines 416-421) sections of the revised manuscript.

Specific comments 12: *Line 345. Please create a Supplementary table with the information on cancer type/subtypes, study design with sample sizes and reference. Please refer to this in the paper along with Fig 1b.*

Response: We have provided **Supplementary Table S1** to present the detailed information on cancer GWAS, including study design, sample size, and references, and referred to it along with **Figure 1B** throughout the revised manuscript (Lines 153, 424-425).

Specific comments 13: *Line 443-444, it seems worth mentioning what is different in this analysis so that you now include non-European descent subjects whereas before they were omitted.*

Response: As we clarified in the responses to the **Overall comments**, all analyses and results presented in the revised manuscript now include data from subjects of all ancestries, in contrast to our initial approach that primarily focused on European descent.

Specific comments 14: *Line 480-482 this sentence is not clear.*

Response: A consistent three-way association requires the association directions of CpG-cancer, CpG-gene, and gene-cancer to be consistent. We evaluated gene-cancer associations through two methods: (1) assessing genetically predicted gene expression in association with cancer risk; (2) comparing gene expression between tumor and adjacent normal tissues. For a trio to be valid, the association direction from either of these methods must correspond with the direction indicated by the CpG-cancer and CpG-gene associations. For example, if a CpG was associated with an increased cancer risk while decreased expression of its target gene, the target gene should be associated with a decreased cancer risk or have a lower expression in tumor tissues than in adjacent normal tissues to make up a valid CpG-gene-cancer trio. This explanation has been elaborated in the revised manuscript (Lines 564-571).

Specific comments 15: *Figure 3D Is there any TCGA data for the boxplots?*

Response: We were unable to conduct differential DNA methylation analyses for ovarian cancer-associated CpGs in TCGA, as only 10 ovarian cancer tissue samples have DNA methylation data. Consequently, a boxplot could not be drawn for **Figure 3D**. We have clarified this in the revised manuscript (**Lines 208-210**) and the **Figure 3 Legend** (**Lines 796-798**).

Special comments 16: Supplemental Table 1 should be provided as a spreadsheet so that the columns are not cutoff due to the page size.

Response: We apologize for any inconvenience caused by the format of **Supplemental Table 1**. To address this issue, we have now provided an Excel file containing all supplementary tables to facilitate the review process.

Special comments 17: Supplemental Table S9. Define tumor subtype. If it's colon and rectal, the word 'sites' is more appropriate than 'subtypes'.

Response: Yes, we meant colon and rectal. We have used "sites" to replace "subtypes" in the footnote of this table (**now Supplementary Table S17**).

Special comments 18: Table S11 footnote, typo. Why use meta-analysis here when earlier tables used subtype adjustment?

Response: We have corrected this typo. We acknowledge that the analyses method should be consistent and have conducted analyses combining lung adenocarcinoma and squamous cell carcinoma samples with tumor subtype information adjusted. These results are presented in **Supplementary Table S19**.

Typos: (1) *Line 82, no s on models*; (2) *Line 86, issue should be tissue*; (3) *Line 100 except not expect, and fix percentages*; (4) *Line 105, exclusive not exclusively*; (5) *Line 109, cancer not caner*; (6) *Line 127, the capability of the approach*; (7) *Line 159, MethylationEPIC and HumanMethylation450 spelling*.

Response: We appreciate the reviewer for pointing out these typos and have corrected them in the revised manuscript: (1) **Line 122**; (2) **Line 125**; (3) this sentence has been deleted; (4) **Line 156**; (5) **Line 168**; (6) **Line 190**; (7) **Line 215-216**.

Responses to Reviewer #2.

Overall comments 1: *Examining the relationship between genotype and DNA methylation is a crucial area of study that can bridge the gap between genotype, phenotype, and diseases. These analyses hold promise for enhancing our understanding of cancer risk by exploring the interplay between genetics and epigenetics. However, the manuscript fails to clarify the conceptual distinctions between genetically influenced CpGs and mQTLs. Additionally, it remains unclear how much this method improves upon canonical QTL-based methods. The lack of adequate benchmarking undermines the credibility of the newly identified disease-relevant CpGs.*

Response: We appreciate the emphasis on distinguishing genetically influenced CpGs from mQTLs. Typically, the mQTL approach focuses on the effect of a single genetic variant that is significantly associated with CpG methylation, whereas our genetic model approach utilizes variable selection algorithms that consider all variants contributing to CpG methylation prediction, irrespective of their individual significance, to improve the prediction performance. We clarified this in the **Discussion** section of the revised manuscript (**Lines 281-284**). To benchmark our method against canonical mQTL-based approaches, we directly compared the prediction performance of our prediction models with mQTLs using identical datasets and the same criteria for model reliability ($R > 0.1$ and $P < 0.05$). In our analysis of seven tissues, we developed reliable prediction models for 478,360 CpGs. In contrast, the single best mQTLs could reliably predict only 33.1% of these CpGs. Moreover, among the CpGs that could be reliably predicted by both approaches, our models exhibited significantly higher accuracy (larger R values) compared to those based on mQTLs (all P values from paired t-test $< 2.20 \times 10^{-16}$; **Supplementary Figure S1**). These results demonstrate the enhanced predictive performance of our approach, providing compelling evidence for the credibility of the disease-relevant CpGs identified in our study. Detailed information on this benchmark has been incorporated in the revised manuscript (**Lines 133-141**).

Further comments 1: *The term 'genetically determined methylation' used in this study is inappropriate and open to debate, as the study examines general associations rather than causality for the identified CpG sites.*

Response: We appreciate the suggestion. In the revised manuscript, we have replaced all instances of “genetically determined methylation” with “genetically predicted methylation”.

Further comments 2: *These genetically associated CpGs were pinpointed using both tissue-specific and whole tissue models. The authors assert that the whole tissue model can leverage information from other tissues to identify tissue specific CpGs, but this reasoning needs explicit clarification.*

Response: In the revised manuscript, we added more details about the “cross-tissue” methods in the **Methods** section (**Lines** 462-464). The Unified Test for Molecular Signatures (UTMOST) method was established by Hu et al. in 2019 (PMID: 30804563). To clarify, “cross-tissue models” in our context refers to tissue-specific models that are enhanced by incorporating genetic effects information from other tissues. This is based on the premise that these models can only leverage information from other tissues when there’s a certain similarity in genetic effects on DNA methylation with the primary tissue in focus. Essentially, while UTMOST models benefit from cross-tissue information, they remain fundamentally tissue-specific.

Further comments 3: *Relevance exploration for genetically determined CpGs involved utilizing summary statistics of susceptibility from different cancers. Although GWAS studies from European ancestry were employed to mitigate differences in genetic architecture, specific effects related to linkage disequilibrium persisted, and determining the likely causal SNPs remained challenging.*

Response: We recognize the challenge in determining causal SNPs, particularly in the context of linkage disequilibrium effects. To address this, we conducted colocalization analyses using the R package *Coloc* for each significant CpG-cancer associations identified in our study to estimate the posterior probability (PP) that a shared causal variant influences both the DNA methylation level of the CpG site and cancer risk. Among the 4,461 significant CpG-cancer pairs, 1,454 (32.6%) showed a moderate probability of colocalization ($PP.H4 > 0.5$) and 866 (19.4%) exhibited a high probability of colocalization ($PP.H4 > 0.8$). Notably, in ovarian cancer, 200 (80.6%) and 117 (47.2%) of 248 significant pairs demonstrated a moderate or high probability of colocalization, respectively. In each colocalization, the variant with the highest $PP.H4$ was considered the most likely causal variant. Detailed descriptions of these analyses and results have been included in the revised manuscript (**Lines** 156-165, 490-494). The full results are presented in **Tables 1-3** and **Supplementary Tables S2-S8**.

Further comments 4: *Moreover, the analysis and result interpretation are challenging to follow because the authors have structured the paper assuming a high level of familiarity with their studies, making it difficult for readers unfamiliar with their work to grasp the content. It would be helpful, for instance, to provide a brief introduction to the software and methods used before delving into result interpretation and presentation. The term "predicted expression" (mentioned in line 183) lacks clarity. Also, the predicted value appeared multiple times and referring to various aspects without clear definition.*

Response: We sincerely appreciate the feedback regarding the clarity and organization of our paper. In response to these concerns, we have incorporated concise introductions to the software and methods used at the start of each relevant section (**Lines** 117-120, 133-136, 146-149, 159-161, 171-175, 194-197, 210-213, 225-226, 232-235, 244-251, 255-256). These introductions are designed to facilitate a more intuitive understanding of our approach, aiding readers in comprehending the results and their interpretation. Regarding the term “predicted expression”, it refers to the genetically predicted expression levels of genes. To improve clarity, we have now ensured that every instance of “predicted value” in the revised manuscript is accompanied by a clear, concise, and context-specific explanation, thus minimizing potential confusions.

Minor comments 1: *The footnotes accompanying the tables and the tables within the PDF file are not comprehensive, leaving important information out.*

Response: We have meticulously revised the footnotes for all tables and supplementary tables in our manuscript. These updated footnotes now include all essential details, ensuring a complete and clear understanding of the data presented in each table.

Minor comments 2: *the explanations regarding long-range associations (those beyond 1Mb) are insufficient.*

Response: We appreciate the suggestion and have added a detailed paragraph to discuss these long-range associations in the **Discussion** section (**Lines** 335-350) of the revised manuscript. The identification of 92 CpGs located >1Mb away from the nearest GWAS-identified cancer risk variants underlines the ability of our CpG-based association test approach to uncover potential novel cancer susceptibility loci, extending the reach of traditional variant-based tests in GWAS. Notably, over 57% and 41% of these 92 CpGs exhibited moderate to high colocalizations with cancer risk, implying their potential as potential causal DNA methylation biomarkers for cancer risk. For instance, the CpG site cg05649751 in the 12q24.11 locus not only showed a significant positive association ($P=2.41 \times 10^{-7}$) and a strong colocalization (PP.H4=0.95) with prostate cancer risk but also had a significant correlation with increased *SH2B3* gene expression, a gene implicated in lung, colorectal, and breast cancer risks (PMID: 26319099). These results suggest a possible role for cg05649751 and *SH2B3* in prostate cancer susceptibility at the 12q24.11 locus, demonstrating the value of our approach in identifying critical loci in cancer susceptibility.

Minor comments 3: *The figure legends lack crucial details. For instance, in Figure 1 (lines 655-659), it is unclear what the range of values represents in panel A, and the meaning of 'n' in panel B is not explained.*

Response: We have updated **Figure 1** legend in the revised manuscript (**Lines** 767-773) to provide more detailed explanations. For panel A of **Figure 1**, the range of values depicted represents the minimum to maximum numbers of results across the seven cancer types studied. The "n" in panel B refers to the number of samples utilized in the development of our DNA methylation prediction models. These clarifications have been included to ensure a comprehensive understanding of the data presented in **Figure 1**.

Reviewers' Comments:

Reviewer #1:

Remarks to the Author:

The authors build over 1 million genetically predicted CpG methylation models that are then used to identify novel cancer risk loci for seven different tissues (breast, colon, kidney, lung, ovary, prostate and testis). The results are combined with gene expression data to try to identify locations where CpG methylation mediates cancer risk through changes in gene expression. Although the data analyzed in this study are not amenable to test the mediation hypothesis, the combination of pairwise correlations do identify new CpG methylation-gene expression associations related to cancer risk and deserving of follow-up studies.

The authors do a good job responding to the previous reviewers. The following points would benefit from further clarification:

Section: "DNA methylation influencing cancer risk through modulating cis-gene expression"

It is confusing to switch from the focus on gene-predicted DNA methylation to the DNA methylation levels directly. Please explain to the reader the reason for this shift in focus.

In addition, after reading the methods I realize these aren't the DNA methylation levels directly but the residuals after covariate adjustment. Please add that detail to the main text.

Line 195: says "...primarily GTEx participants." what non-GTEx participants were included?

Line 244: Please repeat the number of such genes.

Line 257: These 3 associations suggest but does not imply anything. Please replace the word implies. i.e., suggesting that DNA methylation at these CpGs could influence cancer risk through regulating the expression of these genes.

Line 252: How many genes? (in addition to %)

Line 275: I think you mean to say "...with more than 95% being specific to one out of seven particular cancer types."

Discussion, Paragraph 1, last sentence. This is overstated. Although you have reported some interesting associations, this is discovery-based and uses pairwise associations. We have not learned anything about the interplay of genetics, epigenetics and gene expression in cancer etiology.

For TCGA data processing:

Line 510. Please list the TCGA names of the subtypes combined for each cancer site.

Line 511: I am aware of quantile-normalization as a between-sample normalization method. How is this applied within sample? How do you inverse-normalize within-CpG?

Methods, Identifying CpG-gene-cancer Trios

Line 526-527 If I understand correctly, "DNA methylation data involved in prediction model development in the present study" would be more clear if you said "DNA methylation residuals used for prediction model development". If I don't understand, please clarify.

Lines 531-532: "TPM values were quantile-normalized within samples and then inverse-normalized within genes." I have the same questions I had for line 511.

Figure 3 Legend. "OR, odds ratio of genetically predicted DNA methylation or gene expression" Please specify that the OR is for cancer.

Line 798, it seems you are missing a word in "extremely sample size"

Figure 3. Do scatter plots for breast cancer and colorectal cancer show DNA methylation residuals? If

not, why not? It seems residuals are used in Figures 3C & 3D. This needs to be specified in the Figure legend.

Reviewer #2:

Remarks to the Author:

The authors have addressed all of my concerns.

Authors' Responses to Comments from Reviewers

Responses to Reviewer #1.

Overall comments: *The authors build over 1 million genetically predicted CpG methylation models that are then used to identify novel cancer risk loci for seven different tissues (breast, colon, kidney, lung, ovary, prostate and testis). The results are combined with gene expression data to try to identify locations where CpG methylation mediates cancer risk through changes in gene expression. Although the data analyzed in this study are not amenable to test the mediation hypothesis, the combination of pairwise correlations do identify new CpG methylation-gene expression associations related to cancer risk and deserving of follow-up studies.*

The authors do a good job responding to the previous reviewers. The following points would benefit from further clarification:

Comment 1: *Section: “DNA methylation influencing cancer risk through modulating cis-gene expression”*

It is confusing to switch from the focus on gene-predicted DNA methylation to the DNA methylation levels directly. Please explain to the reader the reason for this shift in focus.

Response: We appreciate your suggestion. The shift from genetically predicted DNA methylation to directly measured DNA methylation reflects our prioritization of ground truth data whenever available. In this study, we utilized genetically predicted DNA methylation in our cancer risk analyses due to the challenges of conducting population-based prospective studies with measured DNA methylation data of pre-diagnostic tissue samples from cancer-free subjects. We have added a sentence to clarify this shift: “To accurately estimate the association between DNA methylation and gene expression, directly measured DNA methylation and gene expression data were used” (**Lines 226-228**).

Comment 2: *In addition, after reading the methods I realize these aren't the DNA methylation levels directly but the residuals after covariate adjustment. Please add that detail to the main text.*

Response: We appreciate your attention to this. However, for eQTM analyses presented in this section, we used directly measured DNA methylation levels, not the residuals, as described in the **Methods** section (**Lines 541-546**).

Comment 3: *Line 195: says “...primarily GTEx participants.” what non-GTEx participants were included?*

Response: As we described in **Line 194**, a portion of gene- and splicing-based TWAS results were obtained from two published studies, one for colorectal cancer (ref #4; PMID: 36539618) and the other for breast cancer (ref #26; PMID: 37164006). Both studies utilized GTEx data for prediction model development, with the colorectal cancer study additionally using data from the Study of Colorectal Cancer in Scotland (SOCCS) and the University of Barcelona and University of Virginia (BarcUVa). We have clarified this by adding the citation for the colorectal cancer study (ref #4) at the end of this sentence (**Line 198**).

Comment 4: *Line 244: Please repeat the number of such genes.*

Response: There are 360 genes significantly associated with cancer-associated CpGs in eQTM analyses. We have added this number in this sentence (**Line 248**).

Comment 5: *Line 257: These 3 associations suggest but does not imply anything. Please replace the word implies. i.e., suggesting that DNA methylation at these CpGs could influence cancer risk through regulating the expression of these genes.*

Response: We appreciate your suggestion and have replaced “implying” with “suggesting” (**Line 264**).

Comment 6: *Line 252: How many genes? (in addition to %)*

Response: We have added the number of genes in addition to the percentages for all cancers (**Lines 255-259**).

Comment 7: *Line 275: I think you mean to say “...with more than 95% being specific to one out of seven particular cancer types.”*

Response: Yes. We have replaced “a” with “one out of seven” (**Line 282**).

Comment 8: *Discussion, Paragraph 1, last sentence. This is overstated. Although you have reported some interesting associations, this is discovery-based and uses pairwise associations. We have not learned anything about the interplay of genetics, epigenetics and gene expression in cancer etiology.*

Response: We appreciate your comment and have removed this sentence (**Line 285**).

Comment 9: *For TCGA data processing:*

Line 510. Please list the TCGA names of the subtypes combined for each cancer site.

Response: We have added the TCGA names of the subtypes combined for each cancer site (**Lines 516-518**).

Comment 10: *Line 511: I am aware of quantile-normalization as a between-sample normalization method. How is this applied within sample? How do you inverse-normalize within-CpG?*

Response: We appreciate your highlighting these inaccurate descriptions. We first performed quantile normalization between samples. Then, for each CpG site, we conducted inverse normalization on the methylation values of all samples. We have clarified these in **Lines 518-520**.

Comment 11: *Methods, Identifying CpG-gene-cancer Trios*

Line 526-527 If I understand correctly, “DNA methylation data involved in prediction model development in the present study” would be more clear if you said “DNA methylation residuals used for prediction model development”. If I don’t understand, please clarify.

Response: We appreciate your suggestion. To clarify, DNA methylation residuals were utilized for prediction model development in cancer risk analyses, however, directly measured DNA methylation levels were used for eQTM analyses in this section (**Lines 541-546**; response to **Comment 2**). Both of these data were sourced from the same set of GTEx participants. To provide further clarity, we have revised the sentence as follows: “Directly measured DNA methylation data, which were utilized for residual calculation during prediction model development for cancer risk analyses, along with gene expression data downloaded from GTExPortal, were employed for these analyses” (**Lines 535-538**)

Comment 12: *Lines 531-532: “TPM values were quantile-normalized within samples and then inverse-normalized within genes.” I have the same questions I had for line 511.*

Response: We have revised the description to clarify that TPM values were quantile-normalized between samples and then for each gene, we performed inverse-normalization on expression values of all samples (**Lines 541-543**). We have also revised “**Line 511**” (now **Lines 518-520**) to clarify that DNA methylation β values were quantile-normalized between samples and then for each CpG site, we conducted inverse-normalization on the DNA methylation values of all samples.

Comment 13: *Figure 3 Legend. “OR, odds ratio of genetically predicted DNA methylation or gene expression” Please specify that the OR is for cancer.*

Response: We have clarified that OR represents the odd ratio for cancer risk per SD increase in genetically predicted DNA methylation or gene expression levels in **Figure 3 Legend (Line 804)**.

Comment 14: *Line 798, it seems you are missing a word in “extremely sample size”*

Response: We appreciate your pointing it out. We have added the missing word “small” (**Line 815**).

Comment 14: *Figure 3. Do scatter plots for breast cancer and colorectal cancer show DNA methylation residuals? If not, why not? It seems residuals are used in Figures 3C & 3D. This needs to be specified in the Figure legend.*

Response: We confirm that for all plots in **Figure 3**, directly measured DNA methylation values after quantile-normalization and inverse-normalization, not DNA methylation residuals, were used. This is consistent with the way we perform these eQTM analyses (**Lines 541-546**; responses to **Comment 2 & 11**). We clarified this in the legend of **Figure 3 (Lines 801-803)**.

Responses to Reviewer #2.

Overall comments: *The authors have addressed all of my concerns.*

Comment 1: *The record is available to the public, but accessing files requires users to provide their name and email. I opt not to request access to ensure anonymity prior to publication.*

Response: We appreciate your understanding. The data and code have been made publicly available on March 26, 2024, following the resubmission. They can be accessed through the link provided in the revised manuscript: <https://zenodo.org/records/10810820> (**Line 592**).

Reviewers' Comments:

Reviewer #1:

Remarks to the Author:

The authors have addressed all of my concerns.